# Prediction of biomarkers and therapeutic combinations for anti-PD-1 immunotherapy using the global gene network association

Chia-Chin Wu [1✉], Y. Alan Wang [2✉], J. Andrew Livingston [3,4], Jianhua Zhang [1] & P. Andrew Futreal [1]

Owing to a lack of response to the anti-PD1 therapy for most cancer patients, we develop a network approach to infer genes, pathways, and potential therapeutic combinations that are associated with tumor response to anti-PD1. Here, our prediction identifies genes and pathways known to be associated with anti-PD1, and is further validated by 6 CRISPR gene sets associated with tumor resistance to cytotoxic T cells and targets of the 36 compounds that have been tested in clinical trials for combination treatments with anti-PD1. Integration of our top prediction and TCGA data identifies hundreds of genes whose expression and genetic alterations that could affect response to anti-PD1 in each TCGA cancer type, and the comparison of these genes across cancer types reveals that the tumor immunoregulation associated with response to anti-PD1 would be tissue-specific. In addition, the integration identifies the gene signature to calculate the MHC I association immunoscore (MIAS) that shows a good correlation with patient response to anti-PD1 for 411 melanoma samples complied from 6 cohorts. Furthermore, mapping drug target data to the top genes in our association prediction identifies inhibitors that could potentially enhance tumor response to anti-PD1, such as inhibitors of the encoded proteins of *CDK4*, *GSK3B*, and *PTK2*.

[1] Department of Genomic Medicine, The University of Texas MD Anderson Cancer Center, Houston, TX, USA. [2] Department of Cancer Biology, The University of Texas MD Anderson Cancer Center, Houston, TX, USA. [3] Department of Sarcoma Medical Oncology, The University of Texas MD Anderson Cancer Center, Houston, TX, USA. [4] Department of Pediatrics, The University of Texas MD Anderson Cancer Center, Houston, TX, USA. ✉email: perwu777@gmail.com; yalanwang@mdanderson.org

Breakthroughs in cancer immunotherapies have opened a new front in the war against cancer. Instead of directly targeting cancer cells using specific inhibitors, immunotherapies stimulate and modulate the host's immune system to eliminate cancer cells. Recently, immune checkpoint blockade (ICB), which enhances T-cell activity by inhibiting immunosuppressive checkpoint molecules such as cytotoxic T-lymphocyte-associated antigen 4 (CTLA-4), programmed cell death 1 (PD-1), and programmed cell death protein ligand 1 (PD-L1), has produced durable responses in some cancer patients. Despite these successes, only a subset of cancer patients benefits from these therapies, and rates of response vary widely among cancer types. Therefore, there is a growing need to understand the mechanisms underlying this de novo resistance, to select predictive biomarkers of therapeutic response, and to identify therapeutic targets that could extend the benefits of ICB[1–3].

Ongoing studies show that both cancer-cell-intrinsic and cancer-cell-extrinsic factors contribute to the response to ICB[3]. Initial immune activation requires the expression of neoantigens by cancer cells, which are encoded by somatic nonsynonymous mutations. Many studies have shown that nonsynonymous mutation burden is one of the most important determinants of responsiveness to ICB[4]. However, a considerable number of patients with a high tumor mutation burden (TMB) have poor responses, and a subset of patients with low TMB can respond to ICB. Recently, both patient cohort studies[4–6] and genetically engineered mouse models[7] have shown that cancer cells can utilize their genetic and epigenetic aberrations to influence various aspects of the immune landscape, such as recruitment of immunosuppressive cells into the tumor microenvironment, stimulation of tumor resistance to T-cell attack, and deregulation of immune checkpoint molecule expression. In particular, alterations in or deregulation of multiple pathways in cancer cells, including the MAPK/PTEN/PI3K, WNT/β-catenin, JAK/STAT, interferon-γ, and antigen processing/presentation pathways, have been shown to be associated with resistance to ICB[2,3,5]. In addition, several cancer-cell-extrinsic factors involving the tumor microenvironments, such as T-regulatory cells, myeloid-derived suppressor cells, macrophages[3], and microbes[8], also affect ICB response.

Recently, various omics-based approaches have been undertaken to identify both tumor intrinsic and extrinsic factors which can serve as predictive biomarkers to ICB. First, several genomic factors, such as neoantigen and TMB[9], mismatch repair deficiency[10], and somatic copy-number variation burden[11,12], have been used to predict ICB response. Second, transcription-level data, such as PD-1/PD-L1 expression level[13], immune-cell infiltration profiling based on transcriptional signatures[14,15], and transcriptional immunoscores[16,17], have also been used for response prediction. Third, microbial taxonomic data obtained from analysis of 16S ribosomal RNA[8] can be also used for response prediction. However, none of these approaches considered relationships between tumor genes/pathways and ICB responses. Owing to the different genetic makeup of each tumor cell, the tumor immune landscape differs greatly not only between/within cancer types, but also varies in regions within a given tumor[18]. Thus, elucidation of the underlying cancer gene/pathway-associated molecular mechanisms would facilitate the development of new avenues for personalized immune-intervention strategies. Although some associations between genetic alterations/gene expression deregulation and patients' responses to ICB were explored by recent next-generation sequencing studies, the small sample size of these studies limits their generalizability[6]. Over the past decades, computational network approaches have been widely used for annotating gene functions, identifying novel disease genes, and predicting therapeutic targets[19–21]. Therefore, the network approaches can help to elucidate the genes/pathways associated with ICB responses.

In this work, we develop a network guilt-by-association approach to identify genes and pathways associated with response to anti-PD-1/PD-L1 ICB (hereafter termed anti-PD-1). Integration of top genes in the network association and TCGA data can identify genes whose expression and genetic alterations could affect response to anti-PD1 in each cancer type. The integration can also select the gene signature of a given cancer type to calculate the MHC I association immunoscore (MIAS) to predict patient response to anti-PD1 in the cancer type. At last, the drug target data can be mapped to the top genes in the network association to identify compounds that could enhance tumor response to anti-PD1.

## Results

**Overview of the network-based approach**. The efficacy of anti-PD-1therapy depends on the presentation of neoantigens by major histocompatibility complex (MHC) class I molecules on the surface of cancer cells for surveillance by cytotoxic CD8+ T cells[22]. Thus, we hypothesized that aberrations of any gene that are close to MHC class I genes in the gene network are likely to deregulate the MHC class I antigen processing and presentation pathway (hereafter termed the MHC I pathway) and affect tumor response to anti-PD1. Our proposed network-based approach is illustrated in Fig. 1.

First, the network guilt-by-association method[19,23] was used to calculate functional relatedness (hereafter termed MHC I association score) of each gene with the MHC class I genes, *PDCD1*, and *CD274* (hereafter termed the MHC I pathway genes) on the basis of its relative proximity to these genes in a compiled gene network (see Methods) (Fig. 1a). Aberrations in genes with higher MHC I association scores are more likely to deregulate the MHC I pathway and, thus, to be associated with tumor response to anti-PD-1 therapy.

Second, we considered pathways that enrich genes with high MHC I association scores are also associated with tumor response to anti-PD-1 therapy (Fig. 1a). Gene set enrichment analysis (GSEA)[24], which can evaluate the genes in a pathway for their distribution in an ordered gene list ranked by the MHC I association scores, was used to identifying pathways associated with response to anti-PD-1 therapy (termed GSEA association analysis). However, because the gene network we used in the association prediction contained molecular interactions across different cellular types and statuses, not all of the predicted genes and pathways are associated with the MHC I pathway in a given cancer type. By leveraging gene expression data of samples from a cohort of cancer patients treated with anti-PD-1 therapy, we were able to identify genes and pathways that are both highly functionally associated with the MHC I pathway and deregulated in a given cancer type (Fig. 1b). The deregulation level of an associated pathway can be also determined using GSEA (termed GSEA deregulation analysis). The truncated product method[25] (see Methods) was used to combine the *p* value of each pathway generated from the GSEA association and GSEA deregulation analyses to identify pathways that are associated with response to anti-PD-1 in a given cancer type.

Third, TCGA transcriptomic data that have been used to study the tumor immune infiltration[26] can also be integrated with our network MHC I-association prediction (Fig. 1c) to identify genes and pathways that are associated with tumor response to anti-PD-1 therapy in a given cancer type. We reasoned that the top genes in our MHC I-association prediction, whose expression is significantly correlated with an immune infiltration score (ESTIMATE)[26] in a given cancer type, would be associated with response to anti-PD-1 therapies in that cancer type. Thus, we first determined the correlation of each gene's expression level with

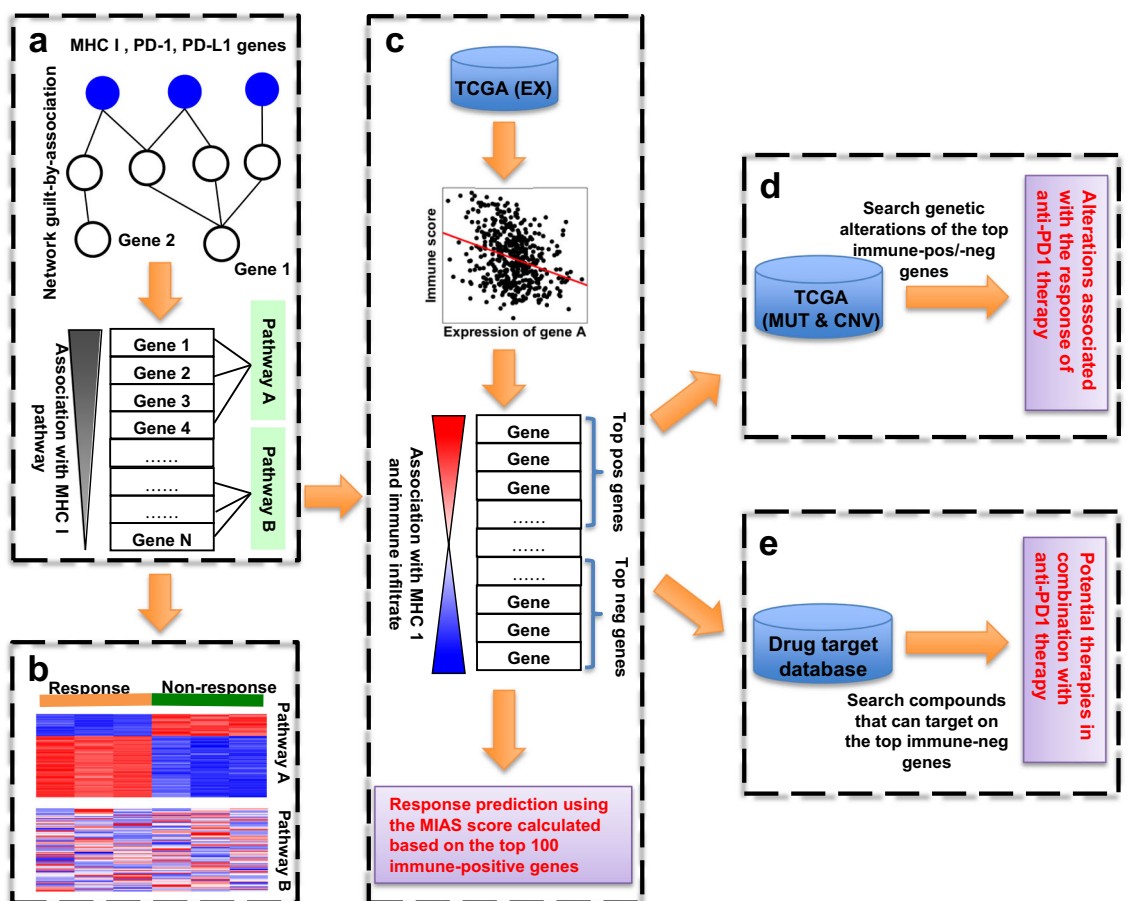

**Fig. 1 Network-based approach. a** Calculation of the functional association score of each gene with MHC I genes (blue node). Genes (e.g., gene 1) that are more functionally associated with the MHC I genes are assumed to be more associated with anti-PD-1 response than other genes (e.g., gene 2) are. Gene set enrichment analysis (GSEA) was used to identify pathways associated with response to anti-PD-1 therapy on the basis of an ordered gene list ranked by the association scores. Pathway A, which enriches genes with higher association scores, is more associated with anti-PD-1 therapy than is pathway B. **b** Gene expression data from tumor samples that were responsive and resistant to anti-PD-1 therapy are leveraged to identify pathways associated with anti-PD-1 therapy response for a specific cancer type or cellular condition. **c** Integration of top genes (e.g., top 10%) in our association predictions (as described in **a**) with the immune correlation of genes (calculated from TCGA transcriptomic data) to select the top immune-positive and -negative genes associated with tumor response to anti-PD-1 therapy in each cancer type. Among them, 100 top immune-positive genes are selected as the signature to predict patient response to anti-PD-1 therapy for a given cancer type. **d** Identification of genomic alterations that would be associated with response to anti-PD-1 therapy in a cancer type by integrating genomic alteration data of a TCGA cancer type with the top immune-positive and -negative genes described in (**c**). **e** Mapping drug target data to the top immune-negative genes (as described in **c**) to identify compounds that can potentially boost response to anti-PD-1 therapy in a specific cancer type. EX gene expression, MUT mutation, CNV copy-number aberration.

the immune infiltration score in a given TCGA cancer type. We then integrated the resulting correlation with the MHC I-association prediction to select top immune-positive and -negative MHC I-associated genes, which are the top 10% genes in our association prediction and whose expression levels were respectively most strongly positively or negatively correlated with the immune infiltration score (absolute correlation ≥ 0.2) in a given cancer type. These genes and their pathways would be associated with tumor response to anti-PD-1 therapy in that cancer type. In addition, we performed a meta-analysis by integration of our association prediction and TCGA transcriptome data (see the Methods) to select the top 100 immune-positive MHC I-associated genes to calculate the signature score, termed the MIAS, which is used to predict patient response to anti-PD-1 therapy for a given cancer type (Fig. 1c). A patient sample with a higher MIAS score would be more likely to respond to anti-PD-1 therapy than those with a lower score. The calculation of the MIAS score is detailed in the Method section.

Fourth, at present, the sizes of clinical cohorts treated with anti-PD-1 therapy are still very limited, making genetic association analyses of response to anti-PD-1 statistically underpowered[6]. Some large-scale cancer genome projects, such as TCGA, have revealed the genomic landscapes of many cancer types. Thus, mining TCGA data using our top MHC I-association prediction would allow us to explore genetic and epigenetic aberrations associated with anti-PD-1 therapy in a given cancer type (Fig. 1d). Here, we selected recurrently altered genes out of the top immune-positive and -negative MHC I-associated genes in a cancer type.

Fifth, we also reasoned that inhibitions of the selected top immune-negative MHC I-associated genes in a given cancer type were likely to be able to boost tumor response to anti-PD-1 therapy for that cancer type. Therefore, drug target data compiled from the DGIdb database[27] were mapped to the top immune-negative MHC I-associated genes in given cancer types, to identify potential inhibitors that could enhance tumor response to anti-PD-1 therapy for that cancer type (Fig. 1e).

**Prediction of genes and pathways associated with MHC I pathway and response to anti-PD-1 therapy**. Our approach first predicted genes that are associated with MHC I pathway genes (Fig. 1a). Some genes that are known to be directly involved in the MHC I pathway, such as *TAP1*, *TAP2*, *CALR*, *TAPBP*, and *B2M*[28], were at the top 1% of our prediction list (Supplementary Data 1a). Alterations of some top genes in our prediction, such as *B2M*, *JAK1*, *JAK2*, *HSP90*, and *IFNG*, are known to be associated with poor response to anti-PD-1 therapy[5,28–31]. Several genes that were recently found to be associated with anti-PD-1 response were also identified in the top list of our association prediction, such as *KRAS*[32], *STK11*[33], *ATR*[34], and *AXL*[35].

Several groups have developed T-cell-based CRISPR/Cas9 screens to identify essential genes associated with the interaction between cancer cells and T cells[36–40]. Therefore, we used the six gene sets from these studies to comprehensively evaluate our predictions. The receiver operating characteristic (ROC) curves and area under the curve (AUC) values for the six gene sets are shown in Fig. 2a. The high AUC values indicated that many of the top genes in our association prediction list would be highly associated with response to anti-PD-1 therapy. By the pairwise overlap analysis, we also found that the top 10% of genes in our prediction significantly overlapped with all six CRISPR-based gene sets (Supplementary Fig. 1). However, the overlap among some CRISPR gene sets from the different studies was not significant. This may result from the use of different techniques or cell sources in their experiments.

Next, we applied GSEA to the ordered list of genes in our association prediction to infer pathways that are associated with response to anti-PD-1 therapy (Fig. 1b). The result also revealed that several pathways known to be associated with response to anti-PD-1 therapy were at the top of our prediction list (Fig. 2c; Supplementary Data 1b): the WNT pathway (the GSEA association plot for this pathway is shown in Fig. 2b), PI3K-AKT pathway, MYC pathway, MAPK pathway, interferon signaling pathway, and TP53 pathway[41–44]. Because these association predictions were not specific to a particular cancer type, we also integrated our predictions with gene expression data of renal cell carcinoma samples from anti-PD-1 therapy responders and nonresponders[45]. We analyzed the gene expression data and also examined the deregulation of these pathways using GSEA. The combination of the GSEA association and GSEA deregulation analyses revealed some pathways that are both highly associated with MHC I pathway in our prediction and significantly deregulated between response and non-response renal cell carcinoma samples (Fig. 2c; Supplementary Data 1b).

In addition, several pathways that have been recently proposed as mechanisms of response or resistance to anti-PD-1 therapy were also identified by our approaches (Supplementary Data 1b), such as DNA damage/repair pathways[46] and defective transcription elongation[47].

**Some cancer genes acting as network hubs are associated with response to anti-PD-1 therapy**. We found that many known cancer genes are in the top list of our MHC I-association prediction, such as *AKT2*, *EGFR*, *TP53*, *PTEN*, *MAPK1*, *MAPK3*, *CTNNB1*, *AKT1*, *JAK1*, and *JAK2* (Supplementary Data 1a). The ROC curve evaluation of our association prediction list using a set of 328 cancer-related genes compiled from the Kyoto Encyclopedia of Genes and Genomes (KEGG) cancer pathways also showed that many cancer pathway genes are in the top list of our prediction (Fig. 2a). The overlap analysis showed that all the six CRISPR-based gene sets also significantly overlapped with the KEGG cancer pathway gene set or with other cancer gene sets compiled from two databases (Fig. 3a). These results indicate that many cancer genes and their related pathways may be strongly associated with the MHC I pathway and patient response to anti-PD-1 therapy.

We reasoned that these cancer genes may act as hubs in the network to directly or indirectly deregulate multiple molecular pathways[19,48], including the MHC I pathway, to suppress the immune system and promote growth, survival, and proliferation of cancer cells. To validate this, we first calculated three kinds of hub centrality scores for all the genes in the compiled network: degree, betweenness, and eigenvector centrality (see the Methods). We then used the six CRISPR-based gene sets to evaluate the results generated by the three centrality methods. The ROC curve in Fig. 3b shows that the results of degree centrality correlated well with the results of all 6 CRISPR gene sets. Similar results were also observed for the other two centrality methods (Supplementary Figs. 2 and 3). Figure 3c, d, and Supplementary Fig. 4, respectively, also show that the results of the degree, betweenness, and eigenvector centrality methods were all highly correlated with our MHC I association prediction. Taken together, these results suggest that some cancer genes act as hubs in molecular networks to indirectly deregulate some pathways, such as the MHC I pathway, to drive immune evasion of cancer cells.

**Mining TCGA data to identify genetic aberrations and pathways associated with anti-PD-1 response**. Some large-scale cancer genome projects, such as TCGA, which have revealed the genomic landscapes of several cancer types, can also be integrated with our network association prediction (Fig. 1c, d) to identify genetic aberrations and pathways that are associated with tumor response to anti-PD-1 therapy in a given cancer type. Among the top 10% of genes in our network association prediction, we first selected the top immune-positive and -negative MHC-I associated genes on the basis of the correlation of their expression level with the ESTIMATE immune infiltration score (absolute correlation ≥ 0.2)[26] across all samples in a TCGA cancer type. We found that the top 10% of our association prediction list (Supplementary Data 1a) significantly enrich genes associated with immune infiltration in most cancer types (Fig. 4a). On average, about 800 top immune-positive and -negative MHC I-associated genes were identified in a cancer type; in total, 2008 genes were identified across all cancer types (Supplementary Data 2). These top immune-positive and -negative MHC I-associated genes are expected to be associated with response or resistance to anti-PD-1 therapy in each cancer type. Pathway analysis of these top immune-positive and -negative genes in each cancer type also revealed several well-known pathways associated with anti-PD1 response in some cancer types, such as the PI3K signaling, IFN-γ, and Wnt pathways (Supplementary Data 3a and 3b).

After comparing the top immune-positive and -negative MHC I-associated genes across all cancer types, we found that only 59 of these genes were shared among all cancer types (Supplementary Data 4). All of the 59 genes were immune-positive and most of them are related to the T-cell-induced cytolytic process. In addition, we also found that the direction of the correlation with immune infiltration for some of these 2008 genes was not consistent across all cancer types; that is, their correlations were positive in some cancer types but negative in others (Fig. 4b). Furthermore, hierarchical clustering of the immune infiltration correlation of the 500 most variable genes of these 2008 genes revealed that cancer types in related tissue lineages were clustered together (Fig. 4b). For example, glioblastoma multiforme and low-grade glioma were clustered together, as were lung squamous cell carcinoma/lung adenocarcinoma; kidney chromophobe carcinoma/kidney renal papillary cell carcinoma; and stomach

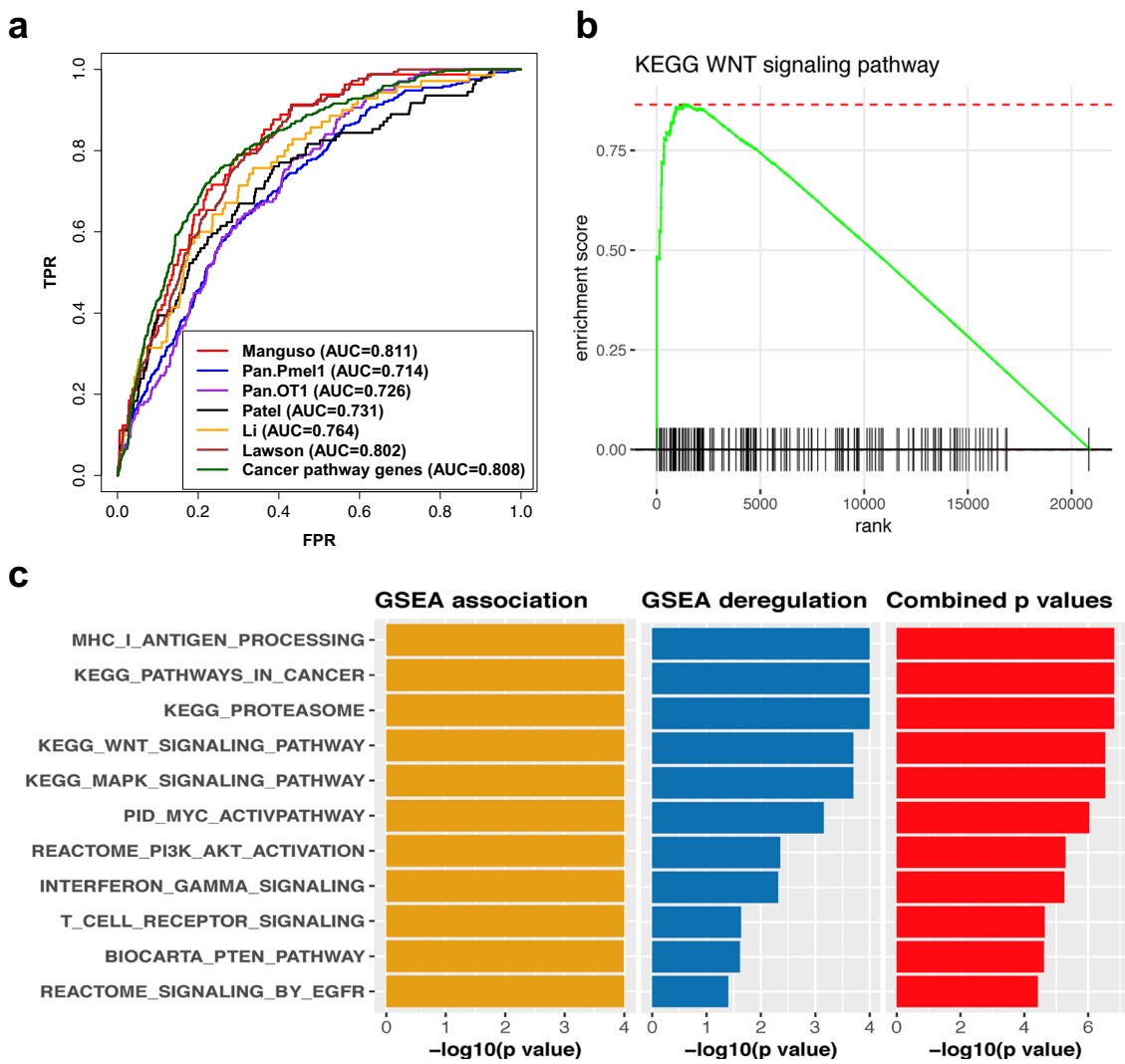

**Fig. 2 Evaluation of the MHC I-association prediction using benchmark gene sets and known pathways associated with anti-PD-1 therapy. a** Receiver operating characteristic (ROC) curves for 6 CRISPR-based gene sets and a set of cancer pathway genes: 81 genes from Manguso (Manguso), 289 genes identified with Pmel-1 T cells from Pan (Pan_Pmel1), 138 genes identified with OT-1 T cells from Pan (Pan_OT1), 109 genes from Patel (Patel), 74 genes from Li (Li), 182 genes from Lawson (Lawson), and 328 genes from the Kyoto Encyclopedia of Genes and Genomes (KEGG) cancer pathways (cancer pathway genes). Genes in these seven gene sets were treated as positive instances, and the remaining genes in the gene network were treated as negative instances. TPR true positive rate, FPR false-positive rate, AUC area under the ROC curve. **b** GSEA association plot of the Wnt pathway from our prediction. GSEA evaluated the genes of the pathway for their distribution in the ordered gene list generated by our association prediction. **c** Some pathways that were significantly associated with the MHC I pathway in our prediction. The length of a bar represents the magnitude of the statistical significance (shown as −log10 value of p-value) of a given pathway in GSEA association analysis, GSEA deregulation analysis, or combination analysis using the truncated product method. The source data of all the panels in this figure are provided in the Source Data file.

adenocarcinoma/stomach and esophageal carcinoma. These results indicate that the molecular mechanisms associated with response or resistance to anti-PD-1 therapy may be tissue and lineage-dependent.

Furthermore, we can map these top immune-positive and -negative MHC I-associated genes to point mutation and copy-number data in TCGA to identify genetic aberrations that may be associated with response or resistance to anti-PD-1 therapy in each cancer type. Table 1 lists some of the identified genetic aberrations in melanoma. Some known genetic aberrations associated with resistance to anti-PD-1 therapy were identified, such as copy-number loss of *B2M* and *IFNGR1*[29,30] and copy-number gain of *MYC*[42]. Several identified genes in Table 1 were recently shown to be associated with adaptive immune response in other cancer types and would be potential biomarkers and therapeutic targets in melanoma. First, the p16-cyclin D-CDK4/

6-retinoblastoma protein pathway is dysregulated in most melanomas[49]. Recently, *CDK4* was found to be associated with antitumor immunity in murine models of breast carcinoma[50], and inhibition of the gene product of *CDK4* can increase PD-L1 expression and enhance the efficacy of anti-PD-1 therapy in vivo[51]. Therefore, *CDK4* copy-number gain would be associated with resistance to anti-PD-1 therapy in melanoma. Second, overexpression of gene *PTK2* has been shown to promote metastasis of human melanoma xenografts[52]. Several studies also showed that silencing of *PTK2* can render pancreatic cancers responsive to anti-PD1[53]. Copy-number gain of *PTK2* would thus be associated with resistance to anti-PD-1 therapy in melanoma. Third, ubiquitination plays important role in the MHC I pathway[54] and in cancer progression[55]. Thus, mutations of ubiquitin C (*UBC*) may be associated with resistance to anti-PD-1 therapy in melanoma. Fourth, some studies have revealed that

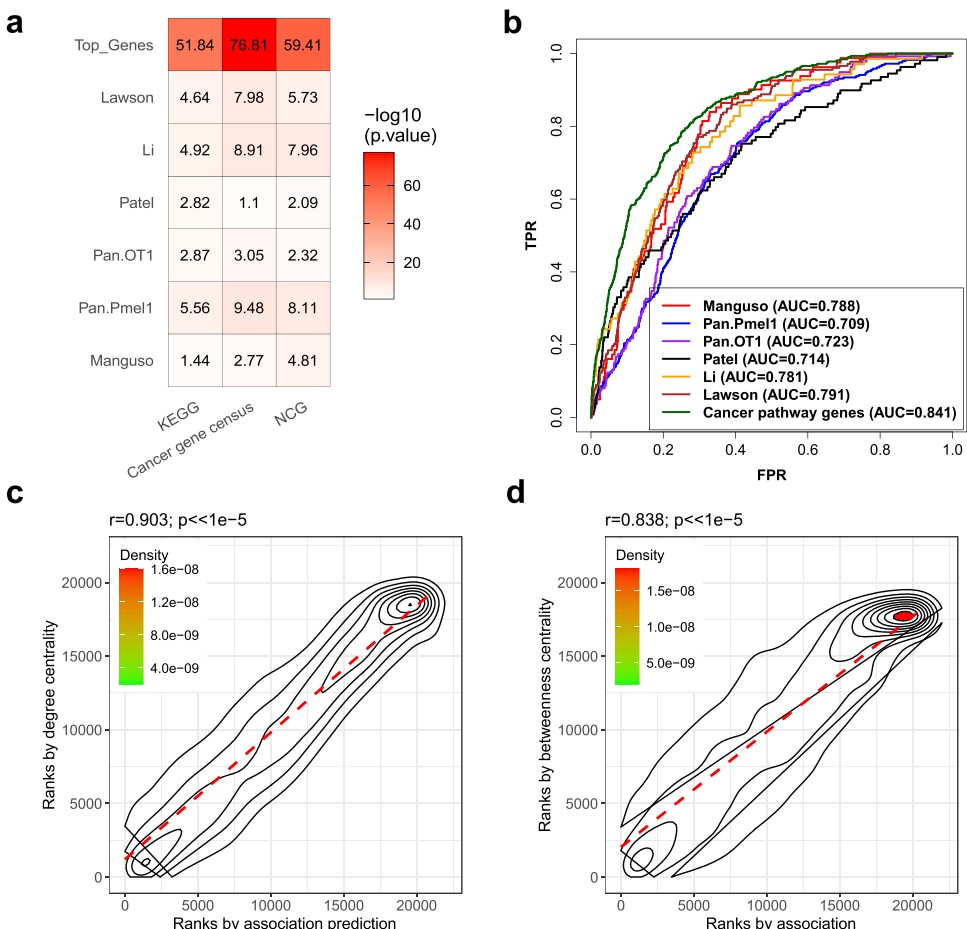

**Fig. 3 Some cancer genes acting as network hubs are associated with response to anti-PD-1 therapy. a** The overlap analysis of the top 10% of genes in our MHC I-association prediction list (Top_Genes) and all the six CRISPR-based gene sets (same denotations as Fig. 2) with the three cancer gene sets compiled from the databases (see Methods). The color scale in the heat map graph indicates the statistical significance of the overlap, −log10($p$-value), calculated by a one-sided hypergeometric test. **b** Evaluation of predictions generated by the degree centrality method using the six CRISPR-based gene sets and the KEGG cancer pathway gene set. The gene sets are indicated as in Fig. 2. **c** Spearman's rank correlation analysis of the prediction generated through MHC I association and the prediction generated using the total degree centrality method. **d** Spearman's rank correlation analysis of the prediction generated through MHC I association and the prediction generated using the betweenness centrality method. Correlation. The correlation coefficient and $p$-value of the analysis is shown in the upper corner of each correlation plot. The source data of all the panels in this figure are provided in the Source Data file.

translation initiation factor 3b (eIF3b), a key subunit of the largest translation initiation factor that acts to ensure the accuracy of translation initiation, is closely related to oncogenesis[56]. Xu et al.[57] reported that *EIF3B* gene expression can accelerate the progression of esophageal squamous cell carcinoma by activating β-catenin signaling, which can promote immune escape and resistance to anti-PD-1 therapy[41]. Thus, a copy-number gain of *EIF3B* may be associated with resistance to anti-PD-1 therapy in melanoma through β-catenin activation.

**Potential therapeutic targets for combination therapy with anti-PD-1 therapy.** We also reasoned that therapies that target some of the top genes and pathways in our association prediction list may overcome resistance and broaden the clinical utility of anti-PD1 therapy. Recently, several clinical trials of targeted therapies and chemotherapies in combinations with anti-PD-1 therapies have been performed. To evaluate our method's ability to predict therapeutic targets for uses in combination with anti-PD-1 therapies, we manually collected 155 target genes of 36 compounds that have been tested in clinical trials or used for combination treatments with anti-PD-1 therapies in cancer

(Supplementary Data 5). As shown in Fig. 5, many of the targets of these compounds are in the top prediction lists of our MHC I-association analysis and the three centrality methods. This indicates that these network approaches can help identify targeted therapeutic strategies to enhance patient response to anti-PD-1 therapies.

We also explored TCGA SKCM data sets to identify additional potential therapeutic targets in melanoma. We reasoned that inhibition of the expression of the selected top immune-negative MHC I-associated genes would enhance immune infiltration, making them potential therapeutic targets for combination treatments with anti-PD-1 therapy. Drug target data compiled from the DGIdb database[27] were then mapped to these top immune-negative MHCI-associated genes to identify potential compounds for use in combination treatment with anti-PD-1 therapy. Table 1 lists some of the identified compounds for SKCM and citations to the literature associating them with immune response. Some of our identified compounds, such as histone deacetylase and poly ADP ribose polymerase inhibitors, which were not included in the 36 compounds used for the aforementioned performance evaluation, are now in ongoing clinical trials in combination with anti-PD-1 therapy[58,59]. Other identified compounds whose targets have been

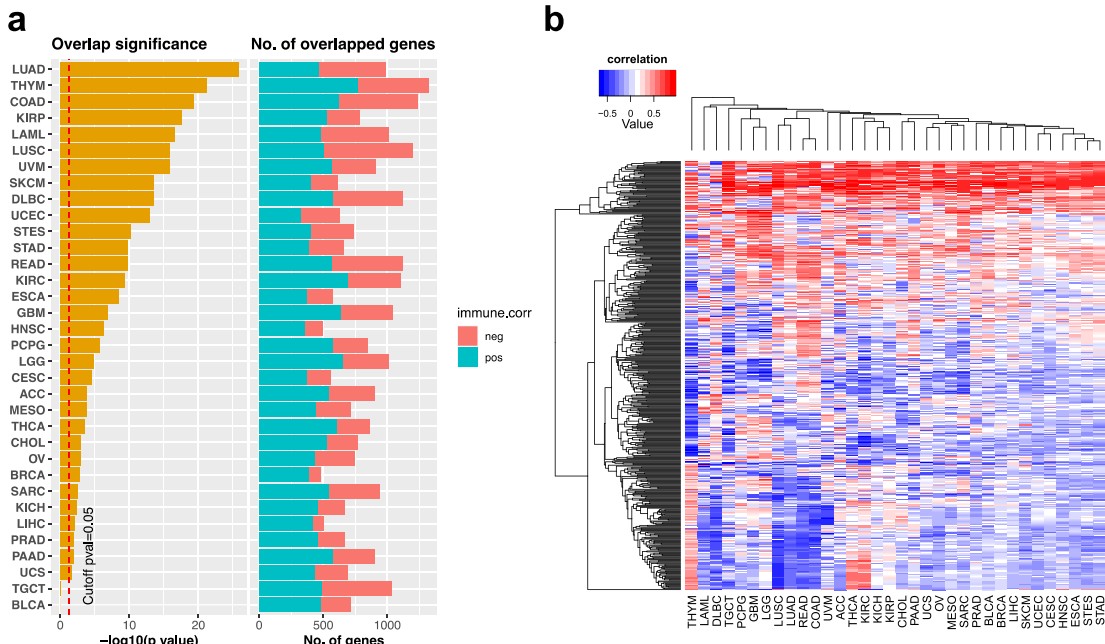

**Fig. 4 Integration of the MHC I-association prediction with TCGA data. a** Overlap analysis between the top 10% of genes in our MHC I-association prediction list and those genes whose expression are significantly correlated with immune infiltration score (absolute correlation ≥ 0.2) for the 34 TCGA cancer types. **b** The hierarchical clustering of cancer types using the 500 top immune-positive and -negative genes with the most variable immune infiltration correlation values across the 34 TCGA cancer types, revealing that cancer types in related tissue lineages were clustered together. The source data of all the panels in this figure are provided in the Source Data file.

**Table 1 Compounds, genetic alterations, and immune-infiltration score correlations of the selected genes identified for TCGA SKCM.**

| Compound | Gene | No. of samples with alterations | Percentile rank (%) in our prediction | Immune-score correlation | Citation |
|---|---|---|---|---|---|
| Denileukin diftitox | *EEF2* | Mut: 3 | 6.6 | −0.259 | 47 |
| Belinostat | *HDAC4* | Mut: 11 | 6.8 | −0.307 | 59 |
| – | *EIF3B* | Amp: 11 | 5.66 | −0.335 | 57 |
| Enzastaurin | *PRKCE* | Mut: 7 | 6.05 | −0.301 | 66 |
| Sulfoxone sodium | *PRMT5* | Mut: 7 | 9.32 | −0.299 | 62 |
| Defactinib | *PTK2* | Amp: 25; Mut: 7 | 3.34 | −0.315 | 53 |
| – | *MYC* | Amp: 24; Mut: 5 | 0.42 | −0.323 | 42 |
| Abemaciclib | *CDK4* | Amp: 14; Mut: 8 | 6.23 | −0.261 | 51 |
| Tideglusib | *GSK3B* | Mut: 3 | 4.12 | −0.257 | 64 |
| Pseudoephedrine | *PRMT1* | – | 0.84 | −0.21 | 61 |
| – | *B2M* | Del: 11; Mut: 5 | 0.39 | 0.719 | 30 |
| – | *IFNGR1* | Del: 11; Mut: 3 | 7.68 | 0.364 | 29 |
| – | *UBC* | Mut: 6 | 0.16 | 0.366 | 54 |

*Abbreviations: Amp, copy-number amplification; Del, deletion; Mut, mutation.*

shown to be associated with the immune response could also be used in combination treatment with anti-PD-1 therapy in melanoma. First, denileukin diftitox, an antineoplastic agent used to treat leukemia and lymphoma, inhibits protein synthesis by ADP ribosylation of elongation factor 2 (eEF2), resulting in cell death[60]. A recent study[47] reported that overexpressed *EEF2* gene suppresses proinflammatory response pathways and correlates with poor response in patients with renal cell carcinoma and metastatic melanoma treated with anti-PD-1 therapy. This implies that denileukin diftitox could be used to enhance the immune response in melanoma. Second, protein arginine methyltransferase 1 (*PRMT1*) is involved in interferon signaling[61], and a few studies have noted that protein arginine methyltransferase 5 (*PRMT5*) may be involved in modulating the expression of some immune response

genes[62]. Several studies recently showed that some epigenetic-targeted drugs, such as DNA methyltransferase inhibitors, can induce T-cell attraction and enhance ICB efficacy in some mouse models[63]. Thus, inhibitors of the encoded proteins of *PRMT1* and *PMRT5* could be used in combination treatments with anti-PD-1 therapy in melanoma. Third, Li et al.[64] showed that glycogen synthase kinase 3β (GSK3β) interacts with PD-L1 and can induce phosphorylation-dependent proteasome degradation of PD-L1. A study by Taylor et al.[65] also reported that GSK3β inhibitors can downregulate PD-1 expression and enhance CD8+ T-cell function in cancer therapy. Thus, the combination of tideglusib, a GSK3β inhibitor, with anti-PD-1 therapy could be a promising strategy for melanoma treatment. Fourth, protein kinase C epsilon gene (*PRKCE*) is overexpressed in most solid tumors and plays critical

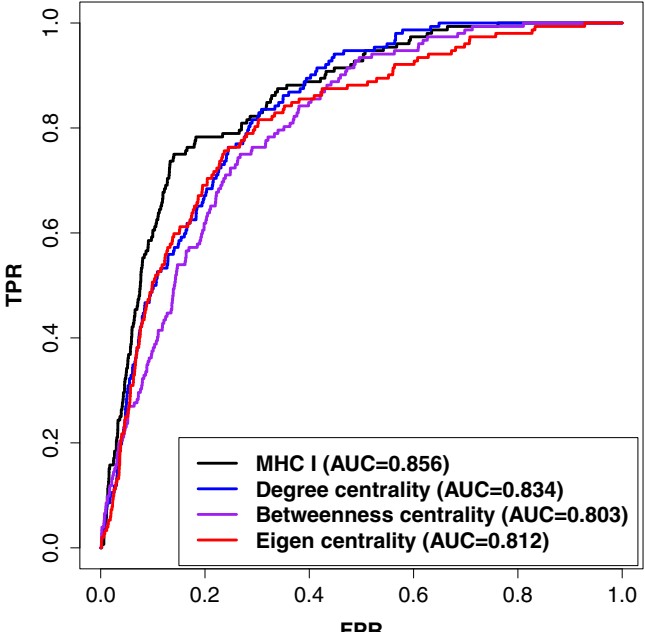

**Fig. 5 Prediction performance of therapeutic targets for combination treatments with anti-PD-1.** ROC curve evaluating the predictions of MHC I-association approach and other network centrality methods using target genes of 36 compounds that have been tested in clinical trials or used for combination treatments with anti-PD-1 therapy. The source data of this figure are provided in the Source Data file.

roles in different cancer-associated pathways, including toll-like receptor 4 (TLR-4) signaling that plays a role in the induction of both innate and adaptive immunity[66]. TLR-4 signaling was recently shown to improve anti-PD-1 therapy during chronic viral infection[67]. Enzastaurin, a protein kinase C inhibitor, thus could be used to enhance anti-PD-1 therapy in melanoma.

**The MHC I-association immunoscore are associated with patient response to anti-PD-1 therapy.** Our MHC I-association prediction can be integrated with TCGA transcriptomic data of a given cancer type to select signature genes to calculate the MIAS score for predicting patient response to anti-PD1 therapies in the cancer type (Fig. 1c). In this work, we demonstrated this capability by applying our approach to melanoma, for which the most anti-PD-1 therapy cohorts are available. We first used a meta-analysis method (see the Methods) to integrate our MHC I-association prediction with the gene expression-immune correlation data, calculated using TCGA melanoma samples, to select the 100 top immune-positive MHC I-associated signature genes (Supplementary Data 6) for response prediction. Herein the TCGA melanoma samples can be considered as the training set for selecting the signature. We then validated the prediction power of this signature using 411 samples compiled from 6 melanoma cohorts (Supplementary Data 7)[16,68–72], in which patients were treated with anti-PD-1 therapy alone or in combination with anti-CTLA-4 therapy. We analyzed the gene expression data of the 411 samples and calculated their MIAS scores using this signature (see Methods). The MIAS scores and the clinical response data of the samples were then used to calculate the AUC to quantify the predictive power of our MIAS approach. This evaluation was applied to each cohort dataset individually as well as the combined dataset from all cohorts. As shown in Fig. 6a, the AUC values of most of the data sets were substantially higher than that by random expectation (AUC = 0.5). However, since the size of some data sets is small, the ROC

curve evaluation may not be reliable[73]. Thus, we also used the Wilcoxon–Mann–Whitney statistic, which is directly connected to the AUC of a ROC curve[74], to evaluate the performance of our MIAS approach. Indeed, we found the results of the Wilcoxon tests and the AUC to be inconsistent in some small data sets; that is, some small data sets had very high AUC values but non-significant p-values on the Wilcoxon test (e.g., the Auslander.PD1.Pre_2018 data set).

We next compared the performance of our MIAS approach with that of three other recently proposed methods, IMPRES[16], TIDE[17], and GEP[75], by the Wilcoxon–Mann–Whitney test (Fig. 6b) and AUC (Supplementary Fig. 5). The results showed that the predictive performance of our MIAS approach was better than those of the three prior methods in most of the datasets. The prediction of MIAS in melanoma was based on the top 100 immune-positive MHC I-associated genes that were selected mainly by the integration of two elements (see the Eq. (8) in Methods): (a) their MHC I association score (denoted MHC-I); (b) the correlation of their expression with the ESTIMATE immune score in melanoma (denoted ESTIMATE). To determine how these two elements of the MIAS approach account for the predictive performance improvement over the three prior methods, we considered four scenarios that select signature genes based on the combinations of the two elements to calculate the ssGSEA enrichment scores (see the Methods) for predicting the response of all the complied 411 melanoma samples. The first scenario denoted MHC-I + ESTIMATE, is our MIAS approach, whose signature genes were selected based on both of the two elements. Thus, the signature genes of the MIAS approach are those top MHC I associated genes that are specifically associated with immune infiltration in melanoma. The second scenario denoted MHC-I + nESTIMATE, selected the top 100 genes only based on their MHC I association score. Accordingly, the 100 MHC I-associated genes in this scenario may not be all specifically associated with immune infiltration in melanoma. The third scenario, denoted nMHC-I + ESTIMATE, selected the top 100 immune-positive genes only based on the ESTIMATE element. The 100 immune-associated genes in this scenario thus may not be all associated with the MHC-I pathway and response to anti-PD1. The fourth scenario, denoted nMHC-I + nESTI-MATE, randomly selected 100 genes for prediction (i.e., neither of the two elements was used to select the genes). We next compared the prediction performance of these four scenarios with those of three prior methods by the Wilcoxon–Mann–Whitney test (Fig. 6c). The result showed that the predictive performance of our MIAS approach (MHC-I + ESTIMATE) was significantly better than those of the three other scenarios, indicating both of the two elements are important to the prediction performance of MIAS. We also found that the third nMHC-I + ESTIMATE scenario has a comparable prediction performance to GEP. Similar to the third scenario whose signature genes are only associated with immune infiltration level, GEP's prediction is based on expression levels of 18 inflammatory genes[75]. Although the signatures used by TIDE in prediction are also related to immune infiltration levels, it has a better performance than the third scenario and GEP, probably due to its two different types of signatures that are respectively associated with dysfunction and exclusion of T cell infiltration[17]. Nevertheless, the prediction performance of TIDE is still much worse than our MIAS approach. One explanation for the better performance of MIAS in anti-PD1 response prediction would be that its signature genes are those top MHC I-associated genes specifically associated with immune infiltration in melanoma.

In addition, as compared to MIAS, TIDE, and GEP, the prediction of IMPRES covers different aspects of immune response mechanisms. The IMPRES prediction is based on the

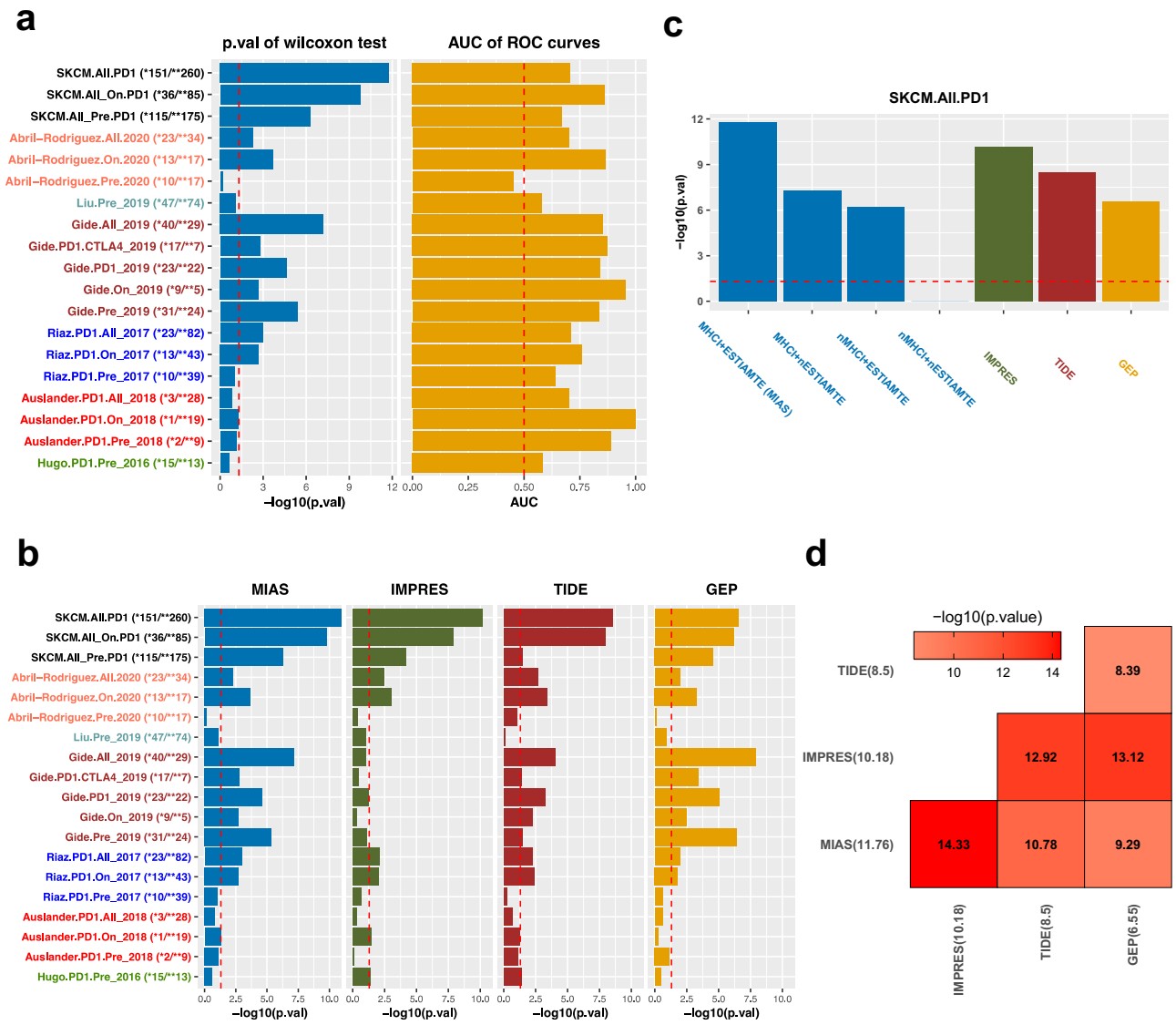

**Fig. 6 Performance of response predictions to anti-PD-1 therapy. a** *p*-values of one-sided Wilcoxon test and area under the curve (AUC) values of receiver operating characteristic (ROC) curves quantifying the predictive performance of our MIAS score for patient response to anti-PD1 across several melanoma patient cohort data sets: Auslander, Gide, Hugo, Liu, Riaz, Abril-Rodriguez, and the aggregated datasets (SKCM.All). "Pre" indicates before treatment; "on", during treatment. The vertical dotted lines in the 2 bar plots respectively represent *p*-value = 0.05 and AUC = 0.5. Datasets of different cohorts are colored with different colors. **b** Performance comparison of our approach with three prior methods, IMPRES, TIDE, and GEP, using one-sided Wilcoxon test across the six melanoma patient cohort data sets. The dotted line in the bar plots represents *p*-value = 0.05. **c** Performance comparison of the four model scenarios with three prior methods using one-sided Wilcoxon test for all the 411 samples merged from all cohorts. The four model scenarios selected different signature genes for response prediction based on the combinations of the two major elements of MIAS (denoted MHC-I and ESTIMATE respectively) of our MIAS approach. The dotted line in the bar plots represents *p*-value of the Wilcoxon test = 0.05. **d** Performance of pairwise integrated predictions of MIAS and the three other methods for all the 411 samples. The color scale in the heat map graph indicates the statistical significance of prediction performance, −log10(*p*-value), calculated by the one-sided Wilcoxon test. The statistical significance of the performance of each individual method was also listed inside parentheses behind its name. The source data of all the panels in this figure are provided in the Source Data file.

15 pairwise relations between the expression of immune checkpoint genes[16], which were selected using the training data of regressing and progressing neuroblastoma but not melanoma patient samples, treated with anti-PD1. Thus, the 15 pairwise transcriptomics relations may not all be predictive for anti-PD1 response in melanoma. Nevertheless, we reasoned that integration of MIAS and IMPRES scores may improve the overall predictive performance because they would be complementary to each other. We evaluated the prediction performance of the integrated scores for the combined dataset (e.g., all the 411 samples) using the Wilcoxon–Mann–Whitney test (Fig. 6d) and AUC

(Supplementary Fig. 6). The result shows that the performance of the integrated score was better than that of the two individual approaches. In contrast, the performance was not improved in the integrated prediction of MIAS and the other two prior methods.

Furthermore, we found MIAS and the three prior methods all performed poorly in pre-treatment datasets, but performed significantly better in on-treatment datasets, except in the Gide data set[71]. To investigate the difference between the pre- and on-treatment samples, we compared their ESTIMATE immune infiltration scores[26] and found that the immune infiltration level in pre-treatment datasets are all significantly lower than in on-

treatment datasets (Supplementary Fig. 7). We also compared the composition of the immune infiltrated cells between all the pre- and on-treatment samples, calculated using ssGSEA enrichment scores of 29 immune cell gene signatures[15], and found that most of the on-treatment samples tend to have more infiltrated immune cells than pre-treatment samples (Supplementary Fig. 8). This is consistent with the expectation that anti-PD1 can boost the immune response. As the aforementioned paragraph, the predictions of MIAS, TIDE, and GEP are based on signature genes associated with immune infiltration. Thus, the poor performances of these methods in pre-treatment datasets are probably due to the low immune infiltration level in these samples. Other types of data, such as mutation burden, may need to be integrated for more robust response prediction in pre-treatment samples.

All the aforementioned results indicate that the MIAS score can be a useful feature to build integrative machine-learning models for response prediction of anti-PD1. The R script for calculating MIAS scores of melanoma samples is available in the GitHub repository (https://github.com/perwu/MIAS). We also used the MIAS and IMPRES scores of the collected 411 melanoma samples (290 pre-treatment and 121 on-treatment) as the data features and applied support vector machine to trained integrative response predictors for pre- and on-treatment melanoma patients samples respectively. The accuracy rates of these predictors calculated by the five fold cross-validation were listed in Supplementary Table 1. These predictors (also available at https://github.com/perwu/MIAS) can help people to predict responses of SKCM patient samples directly using their transcriptomic data.

## Discussion

Our MHC I-association network approaches successfully identified genes and pathways known to be associated with anti-PD1 response. The prediction was also comprehensively validated by several independent benchmark gene sets, including 6 CRISPR gene sets associated with tumor resistance to cytotoxic T cells and targets of the 36 compounds that have been tested in clinical trials for combination treatments with anti-PD1. In addition, target genes of several compounds that were recently shown to be able to enhance tumor response to anti-PD1 are also on the top list of our association prediction (Supplementary Data 1a), such as inhibitors of the encoded proteins of *GSK3B*, *CDK4*, and *PTK2*. In particular, several very recent studies also cross-validated our prediction. First, Litchfield et al.[76] analyzed genomic data of up to 1000 ICB-treated patient samples collected from 12 published cohorts across 7 tumor types, and showed that copy number loss of *TRAF2* is associated with response and *CCND1* amplification is associated with resistance. *TRAF2* and *CCND1* are respectively in the top 4.42% and 6.5% in our association prediction list. Second, *SETDB1* (top 15.13% in our prediction) was found to be associated with MHC I presentation and CD8+ T cell recognition of transposable element-encoded antigens[77]. Amplification of *SETDB1* in human tumors is associated with immune exclusion and resistance to ICB, and *SETDB1* loss can sensitize tumors to anti-PD1[77]. Third, belapectin, an inhibitor of the encoded protein of *LGALS3* (top 8.78% in our prediction), can enhance the clinical and immunological effects of anti-PD1[78]. All these demonstrated that our approach can effectively identify genes and pathways associated with response to anti-PD-1 therapy in cancer.

CRISPR/Cas9 and RNA interference screens recently have been applied to identify genes associated with response to immunotherapy without using any prior biological information. However, when no prior biological information is used, CRISPR/Cas9 and RNA interference screens must be used to assess all

genes. In contrast, by integrating our prediction, we only need to screen the top predicted genes. Our analysis also showed that the limited overlaps among results of some CRISPR-based screens may be associated with different techniques or cell sources used. In contrast, the top genes in our prediction significantly overlapped with all the CRISPR-based gene sets, suggesting that our approach may be relatively unbiased to the CRISPR-based approaches.

Our centrality analysis revealed that some cancer genes act like hubs in their molecular networks and are able to drive immune evasion through directly and indirectly deregulating the MHC I pathways. Therefore, some network centrality methods are also able to be used to identify genes and pathways associated with response and resistance to anti-PD1. The results of an in vivo study by Lesterhuis et al.[79] support this view. Lesterhuis and colleagues used analysis of gene networks inferred from gene expression data of responding and nonresponding tumors in murine models to identify hub genes and modules associated with response to anti- CTLA-4. They showed that targeting some of these identified hub genes with selected drugs dramatically enhanced the efficacy of CTLA-4 blockade in their murine models. However, the targets identified in mice may not be applicable in human tumors, and co-expression networks constructed from small samples may contain many false positives and negatives[80]. Nonetheless, this evidence shows that mutations or deregulation of some hub genes in the gene network can contribute to resistance to immunotherapies; importantly, many of these genes and their pathways are targetable. However, hub genes are normally engaged in multiple other pathways that govern normal cell functions. Thus, to identify hub genes as therapeutic targets that are specifically associated with anti-PD1 in cancer, the hub genes predicted by network approaches still need to be prioritized using other data, such as differential gene expression data between tumor and normal samples, or gene expression data of samples from cancer patients treated with anti-PD-1 therapy (Fig. 1b) or immune infiltration correlation data in cancer (Fig. 1c).

Targeted therapies are directed against important dysregulated molecular pathways or mutant proteins that are required for the growth and survival of malignant cells. There has been a growing interest in combining target therapies with ICB due to the immune-modulating effects of many targeted agents. First, targeted therapies can cause tumor death, which leads to the release of neoantigens and thus enhances the efficacy of ICB[81]. Second, targeted therapies can restore deregulated pathways associated with immune-suppressive mechanisms, such as T-cell dysfunction and exclusion, to overcome resistance and broaden the clinical utility of ICB. In a recent study addressing the first point, Colli et al.[81] estimated the proportion of solid tumors that might benefit from immuno-targeted combination therapy. They surveyed thousands of TCGA genomic profiles for cases with specific mutations targeted by current agents and with a burden of nonsynonymous mutations that exceeded a proposed threshold for response to ICB. Unlike the work of Colli et al., our approach addresses the second point. Our prediction method also can be easily integrated with patient mutation-burden data to identify potential therapeutic targets and compounds that comply with both of the two aforementioned clinical benefits for immuno-targeted combination therapies.

Recently, several efforts tested possible drug combinations with checkpoint inhibitors. However, tests of all the possible drug combinations with PD-1/PD-L1 inhibitors may exceed the number of eligible cancer patients who can be enrolled in clinical trials and are therefore not feasible in the context of clinical trials[33]. With the incorporation of drug-target information, our approach can be used to effectively identify compounds that may

have action in pathways that enhance response to anti-PD-1 therapy. However, precisely predicting the possible therapeutic effects of the identified drugs in the combination treatments is difficult for several reasons, including target-dependent drug-binding affinities, unclear mechanisms of drug action, and different types of phenotypic effects (positive or negative) induced by different targets of a drug[19]. However, our approach could serve as a quick way to initially screen a small number of potential drugs for further evaluation and to help reduce the search space in screenings[82]. Moreover, our approach may help to reduce drug discovery costs by finding new uses of some approved drugs in the combination treatment with anti-PD1 for cancer patients (i.e., drug repurposing).

Our integrative analysis of TCGA data and our network association prediction (Supplementary Data 1a) also indicated that the molecular mechanisms associated with patient response to anti-PD-1 therapy are context-dependent and tissue-specific. Other studies also showed that immunoregulation are tissue specific[83]. Therefore, integrating other high-throughput data, such as gene expression data, with the top genes in our association prediction list (Supplementary Data 1a) would further help prioritize important biomarkers and potential therapeutic targets for combination treatments with anti-PD-1 therapy for a given cancer type. Furthermore, integrating our predictions with individual patient data (genomic, transcriptomic, or epigenetic) may help to develop personalized immuno-oncology treatments.

Integration of our MHC I-association prediction with TCGA transcriptomic data of a given cancer type can select signature genes for calculating the MIAS score to predict patient response to anti-PD1 of that cancer type. In this work, we demonstrated this capability in melanoma. We selected 100 top immune-positive MHC I-associated genes as the gene signature in melanoma using TCGA SKCM transcriptomic data and used the 411 melanoma samples complied from six cohorts to validate the prediction performance of the MIAS score calculated by the signature genes. Our MIAS approach also could be applied to other cancer types to select the signature genes for predicting patient response in these cancer types. However, factors in addition to the deregulation of the MHC I pathway, which have been shown to be associated with response/resistance to anti-PD1, such as mutation burden, the microbiome, environmental factors, germline genetics, and immune infiltration, also need to be included in the response prediction[84]. Thus, the development of integrative computational approaches on the basis of different cancer cell-intrinsic and -extrinsic factors would perform better than approaches that only consider any individual factor. For instance, the combination of mutation burden and tumor aneuploidy scores was shown to better predict response to ICB than either score alone[11,12]. Our analysis also showed that the integration of the MIAS and IMPRES scores also have a better prediction performance than the two individual methods.

In addition, our results also showed that MIAS, IMRPES, and TIDE all performed significantly worse in pre-treatment cohorts than in on-treatment cohorts, and the estimated immune infiltration levels in pre-treatment datasets are also significantly lower than that in on-treatment datasets. This suggests that anti-PD1 therapy may induce dramatic transcriptome changes in the tumor microenvironment and thus increase the prediction power of the three methods in on-treatment samples. Similarly, Chen et al.[85] found that gene expression profiles between responder and nonresponder are not significantly different for pre-treatment samples, but much more significantly for on-treatment samples. The gene expression of antigen presentation, T-cell activation, and T-cell homing are significantly upregulated in responders of those on-treatment samples. They concluded that gene expression profile in early on-treatment samples is highly predictive of

response to anti-PD1 therapy, but is not robust in those pre-treatment samples. Therefore, we suggest that other types of data, such as mutation burden, may need to be integrated for more robust response prediction in pre-treatment samples.

Our analysis also showed that some small data sets had very high AUC values but nonsignificant $p$ values on the Wilcoxon test, suggesting a large patient data set is essential to build and evaluate a model approach robustly. However, it is still challenging to build a comprehensive prediction model by integrating various kinds of factors, mainly owing to a lack of large-scale patient cohorts with multilevel factor data[84]. Furthermore, our response predictor was built based on analysis of bulk gene expression data that mainly reflect the averaged gene expression across different types of cells, including malignant cells, normal host cells, stroma cells, a wide variety of immune cells, and other cells. Single-cell RNA-seq analysis enables the identification and quantification of cell populations present in a given dataset. Therefore, integrating single-cell RNA-seq data and our association network approach may be able to analyze gene expression of cancer cells and tumor cells separately and develop more robust response predictors.

While our approach cannot be used to identify biomarker genes whose molecular interactions have not yet been well characterized because it is based on incomplete molecular interaction data, it will improve overtime as more data are generated. Regardless, these results suggest that our network approach is an effective method to identify genes/pathways associated with response/resistance to anti-PD1 therapy and can be employed as in-silico screening of potential drugs for combination regimens with anti-PD1 therapy in cancer. In addition, some other network approaches, such as network centrality, direct neighborhood relationship, and shortest distance[19,86], also can be used to identify genes and pathways associated with anti-PD1.

## Methods

**Network guilt-by-association using random walk with restart**. A gene network that was compiled from several curated databases[21], consisting of 20,909 genes and over 645,406 molecular interactions, was used in the network guilt-by-association analysis. Given a set of MHC class I genes as bait genes, we used the random walk with a restart to calculate association scores for each gene with the MHC I pathway genes. In a network with n nodes (i.e., $n$ genes in the gene network), the random walk with restart is defined as[23]

$$\mathbf{p_{t+1}} = (1 - \gamma)\mathbf{W}\mathbf{p_t} + \gamma\mathbf{p_0} \quad (1)$$

where $\mathbf{p_0}$ is the initial probability vector in which equal probabilities are assigned to the starting nodes (i.e., MHC class I genes, PD-1, and PD-L1); $\mathbf{p_t}$ is the probability vector containing the probabilities of the nodes at step $t$; γ is the restarting probability; and $\mathbf{W}$ is the transition matrix, which is a column-normalized adjacency matrix of the network. Starting from the set of nodes in the network, the walker will iteratively move from the current nodes to randomly selected neighbor nodes or return to the starting nodes. When iteratively reaching stability (i.e., when the change between $\mathbf{p_t}$ and $\mathbf{p_{t+1}}$ is below $10^{-30}$), the probability vector can present the association scores of all genes in the network with the starting genes. Thus, genes with higher association scores are more functionally associated with the MHC I pathway.

**Centrality measurements to identify hub genes**. Network centrality measures have been used to determine hub genes in a network. These measurements also have also been used to identify genes associated with cancer and other diseases[19]. In this work, 3 centrality measures[87]—total degree, betweenness, and eigenvector centrality—were used to identify genes associated with anti-PD-1 response.

In a binary network (e.g., the curated gene network mentioned above) with $n$ nodes (i.e., a node represents a gene in the curated gene network), the total degree centrality of a node $i$ ($d_i$) is defined as the total number of linkages connecting it with other nodes in the network:

$$d_i = \sum_{i \neq j} a_{i,j} \quad (2)$$

where $d_i$ is the centrality measurement of gene $i$, and $a_{i,j}$ indicates a connection between gene $i$ and gene $j$ in the binary network (i.e., $a_{i,j}$ 1 if there is a linkage between nodes $i$ and $j$ and $a_{i,j} = 0$ otherwise). Total degree centrality is the simplest way to define the centrality of a node in a network, but it only counts the local

impact of a node through its direct connections in a network. Thus, some bottleneck hubs[88], which have few connections with other nodes but act as key connectors in a network, cannot be identified using total degree centrality. Thus, we also used betweenness centrality[87] in the analysis, since it can count the global importance of a node in a network through the node's direct and indirect connections. Betweenness centrality estimates the centrality of a node in a network by considering the fraction of shortest paths that pass through that node. The betweenness centrality of a node $i$ ($b_i$) is defined as

$$b_i = \sum_{s \neq t \neq i} \frac{P_{st}(i)}{P_{st}} \tag{3}$$

where $P_{st}$ is a total number of shortest paths between nodes $s$ and $t$, and $P_{st}(i)$ is the number of those shortest paths between $s$ and $t$ that pass through the $i$.

Another global centrality measurement used in this work is eigenvector centrality, which estimates the centrality of a node in a network based on the concept that nodes are important if they are connected to nodes that are themselves central within the network. The eigenvector centrality score of a node $i$ ($e_i$) is defined as the sum of the centrality values of the nodes that it is connected to[87]

$$e_i = \frac{1}{\lambda} \sum a_{i,j} e_j \tag{4}$$

The equation can be rewritten in vector notation as the eigenvector equation

$$\mathbf{AE} = \lambda \mathbf{E} \tag{5}$$

where $\mathbf{A} = (a_{i,j})$ is the adjacency matrix of the network (i.e., $a_{i,j} = 1$ if node $i$ is connected to node $j$, and $a_{i,j} = 0$ otherwise), $\mathbf{E} = (e_i)$ is a positive eigenvector, and $\lambda$ is an eigenvalue constant.

**Identifying pathways associated with the anti-PD-1 response in a given cancer type**. The truncated product method[25] was used to combine the $p$ values of each pathway generated from the GSEA association and GSEA deregulation analyses to identify pathways that were both highly associated with the MHC I pathway in our prediction and deregulated in a specific cellular condition. In the truncated product method, the product score $W$ of the 2$p$ values ($_{pi}$) that do not exceed a fixed $\tau$ value ($\tau$ was set to 0.01 for both GSEA association and deregulation analysis) can be calculated as

$$w = \prod_{i=1}^{2} p_i^{I(p_i \leq \tau)} \tag{6}$$

where I(.) is the indicator function. The probability of $W$ for $w < 1$ can be evaluated by conditioning on $k$, the number of $_{pi}$ values less than $\tau$:

$$\Pr(W \leq w) = \sum_{k=1}^{2} \Pr\binom{2}{k}(1-\tau)^{2-k}\left(w\sum_{s=0}^{k-1}\frac{(k\ln\tau - \ln w)^s}{s!}\right)I(w \leq \tau^k) + \tau^k I(w > \tau^k) \tag{7}$$

**Gene sets for performance evaluation of the network association prediction**. Two types of gene sets were used to evaluate our predictions. First, T-cell-based CRISPR/Cas9 screens were recently applied to identify genes associated with mechanisms of tumor-cell resistance to killing by cytotoxic T cells. Six gene sets compiled from 5 such studies were used for evaluating our prediction, including 81 genes from Manguso's studies (the $p$ values in the studies $\leq 0.05$)[36], 109 genes from Patel's studies (the $p$ values identified by the RIGER metric in the studies $\leq 0.0005$)[37], 289 genes identified with Pmel-1 T cells and 138 genes identified with OT-I T cells from Pan's studies (the $p$ values in the studies $\leq 0.05$)[38], 74 genes from Li's studies (the $p$ values in the studies $\leq 0.05$)[39], and 182 genes from Lawson's studies[40]. Second, several sets of cancer-associated genes were used to evaluate the association between cancer pathways and our predictions: 328 cancer-related genes compiled from KEGG cancer pathways, 723 genes from the Cancer Gene Census[89], and 2372 genes from The Network of Cancer Genes (NCG)[90]. Fourth, to evaluate our prediction of therapeutic targets associated with anti-PD-1 therapies, we manually collected 155 target genes of 36 compounds that have been in clinical trials and already used for immuno (anti-PD-1)-targeted combination therapies in cancer treatment (Supplementary Data 5). The 36 compounds were collected from several review papers[1,91,92]. The known target genes of these compounds were compiled from the DGIdb database[27].

**TCGA data**. Gene expression data for 34 TCGA cancer types were downloaded from the Broad GDAC Firehose (https://gdac.broadinstitute.org/) and were standardized across samples by quantile normalization. The immune infiltration score of each TCGA sample was calculated using ESTIMATE[26], and the correlation of each gene's expression level with the immune infiltration score across samples of a TCGA cancer type was then calculated. The resulting correlations were integrated with the MHC I-association prediction (Supplementary Data 1a) to select top immune-positive and -negative MHC I-associated genes in a given cancer type (Supplementary Data 2), which are the top 10% genes in the association prediction and whose expression levels were respectively most strongly positively or negatively correlated with the immune infiltration score (absolute correlation $\geq 0.2$) in the cancer type. Mutation calls and copy number segments of the 34 TCGA cancer

types were also downloaded from the GDAC. The R package, CNTools[93] was used to convert the segment data into a copy number of genes. For assessing genes with copy number aberrations, log2 values of copy number $> 0.5$ were considered gains while log2 values $< -0.5$ were considered losses.

**Prediction of anti-PD-1 therapy response and the performance evaluation**. We found most of the top immune-positive genes that are related to T cell-inflamed cytolytic processes (Supplementary Data 3a), similar to the previous studies[75], thus can be direct indicators of T cell-inflamed immune response. However, those top immune-negative genes that are related to diverse pathways associated with cancer progression and immune suppression (Supplementary Data 3b) would be indirect indicators associated with T cell-inflamed immune response. For example, high expression of the top immune-negative genes would be related to high immune suppression, but low expression of these genes are not always associated with high T cell-inflamed immune response because other factors, such as neoantigen burden, also impact immune response. Therefore, we used the following method to select the top 100 immune-positive MHC I-associated genes as the gene signature for predicting patient response to anti-PD-1 therapies in a specific cancer type (Fig. 1c). For each gene $g$, the rank product statistics[94] was applied to merge its rank in the MHC I-association prediction ($r1_g$) and its rank based on the ESTIMATE[26] immune infiltrate correlation ($r2_g$) to have a merged rank score (MR$_g$)

$$MR_g = \left(r1_g \times r2_g\right)^{1/2} \tag{8}$$

Thus, genes with the top merged rank scores, MR$_g$, are the top genes in our MHC I-association prediction and whose expression levels were also most strongly positively correlated with the immune infiltration score in a given cancer type. We then reasoned that the higher the expression levels of the top 100 genes are in a patient sample, the more immune infiltration the sample may have, and the more likely it can respond to anti-PD1 therapy. Therefore, the MIAS of a patient sample $s$ is then defined as

$$MIAS_s = NES_s \tag{9}$$

where NES$_s$ is the ssGSEA[95] normalized enrichment score of the top 100 genes for the sample $s$. A patient sample with a higher MIAS score would be more likely to have a response to anti-PD-1 therapy than would one with a lower score.

In this work, we applied this method to 411 pre- or on-treatment tumor samples (anti-PD-1 or combination of anti-PD-1 and anti-CTLA-4), compiled from 6 melanoma cohorts[16,68–72]. The 100 signature genes (Supplementary Data 6) were first selected by the integrative analysis of TCGA SKCM transcriptomic data and our MHC I-association prediction. The raw count of the RNA-seq data of the 411 samples was first converted to transcripts per million (TPM) that normalize counts for library size and gene length, and then used to calculate the MIAS score of each sample based on the 100 signature genes. To compare the MIAS score across the samples in different cohorts, we selected the 14,820 common genes covered by all the platforms (including the TCGA platform) for the MIAS score calculation. According to the clinical RECIST response information provided in the 6 cohort studies, samples were categorized as a response (complete response and partial response) and non-response (stable disease and progressive disease). Supplementary Data 7 summarizes the response annotations of the samples. Finally, the prediction performance of the MIAS score was performed using the Wilcoxon signed-rank test and AUC of the ROC curve.

We also compared the performance of our method with that of 3 other recently proposed methods, IMPRES[16], TIDE[17], and GEP[75], using the data of the 411 melanoma samples. The IMPRES score of a sample was calculated by counting the number of the identified 15 pairwise gene expression relationships that are fulfilled. The TIDE score of a sample was calculated using a published online tool (http://tide.dfci.harvard.edu/), and the TPM values used for the calculation were further normalized as suggested in the paper. The T cell-inflamed GEP score of a sample was calculated as the ssGSEA[95] normalized enrichment score of the 18 inflammatory genes[75] for the sample. The rank product statistics[94] was used to integrate the predictions of our approach and the other 3 methods.

**Reporting summary**. Further information on research design is available in the Nature Research Reporting Summary linked to this article.

## Data availability
The TCGA genomic and transcriptomic data used in this study are from the Broad Institute's Firehose data portal (https://gdac.broadinstitute.org/). The RNA-seq data of the 411 melanoma samples complied from the 6 published patient cohorts: Auslander[16] (available in Gene Expression Omnibus: GSE115821), Hugo[68] (available in Gene Expression Omnibus: GSE78220), Liu[69] (available in dbGaP: phs000452.v3.p1), Riaz[70] (available in Gene Expression Omnibus: GSE91061), Gide[71] (available in European Nucleotide Archive: PRJEB23709), and Abril-Rodriguez[72] (available in dbGaP: phs001919.v1.p1). The known target genes of compounds were compiled from the DGIdb database (https://www.dgidb.org/). The data underlying the figures are provided as a Source Data file. All the other data supporting the findings of this study are available within the article and its supplementary dataset and information files. Source data are provided with this paper.

## Code availability

The R script for calculating MIAS scores of melanoma samples is available in the GitHub repository (https://github.com/perwu/MIAS)[96] and the corresponding DOI is as follows: https://doi.org/10.5281/zenodo.5715057. The trained anti-PD1 response predictors for pre- and on-treatment melanoma patient samples are also available in the repository.

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

## Acknowledgements

This work was supported by Amschwand Sarcoma Cancer Foundation Award 2017 (C.C.W. and P.A.F.), Cancer Prevention Research Institute of Texas R120501 (P.A.F.), and Welch Foundation's Robert A. Welch Distinguished University Chair Award G-0040 (P.A.F.). This work was also in part supported by NIH 1RO1CA231349-01-A1 (Y.A.W.) Department of Defense PC190813 (Y.A.W.) and Emerson Collective (Y.A.W.).

## Author contributions

This study was conceived of and led by C.W. C.W. designed the study and algorithm. C.W. implemented all the data analysis with input from Y.A.W., J.Z., and P.A.F. C.W., Y.A.W., and J.A.L. wrote the paper. J.Z., and P.A.F. supervised the project. All authors read and approved the final paper.

## Competing interests

The authors declare no competing interests.
