## [Peer Review File · Nature Communications]

Prediction of biomarkers and therapeutic combinations for anti-PD-1 immunotherapy using the global gene network associationReviewers' Comments:

Reviewer #1:

Remarks to the Author:

-- In this manuscript, the authors use a network-based approach to develop predictive methods that can be used identify genetic features, pathways, and therapeutic combinations associated with tumor response to anti-PD1 therapies. This is an interesting study that builds on other studies focused on molecular network-based analyses of complex disease mechanisms and therapeutic target/pathway identification.

-- On page 3, the authors should point out that the tumor immune landscape also varies within a given tumor in an individual patient. This point also raises the important concern of sampling variability and its consequences for the soundness of statistical inference. Do the authors have access to (even limited) data sets from patients in whom repeated sampling was obtained, and to what extent does the MIAS score vary within such samples?

-- The authors should identify the elements of their approach that account for the modest improvement over prior methods (IMPRES, TIDE, GEP), and do so quantitatively (rather than qualitatively, as on p. 12).

-- The notion of hubs as therapeutic targets (associated with anti-PD1 therapy) has significant potential shortcomings, given that by their highly connected nature, hub proteins are engaged in multiple other pathways that govern normal cell function. This important issue should be discussed.

-- The predictions of combination therapies that should enhance anti-PD1 therapy is most interesting and should be tested in vitro (at the very least), especially for agents that have not yet been in clinical trials.

-- The pre- and on-treatment response analysis is informative, but has been insufficiently analyzed. Specially, in a validation set, one should be able to use the pre-treatment analysis to predict the on-treatment changes in gene expression, pathway modulation by treatment, and MIAS score.

-- The random walker approach to network proximity ('guilt-by-association') is one of several possible methods that can be used for the initial network analysis. The authors should consider (at least in the Discussion) some of the others that have been used (simple proximity, rather than 'diffusion' via the walker; neural network-based AI; etc.).

Reviewer #2:

Remarks to the Author:

This is an interesting article based on the re-analysis of publicly available sequencing data of biopsies of patients with cancer treated with immune checkpoint blockade therapies. The strength is the combined analysis of over 350 biopsies, allowing the comparison of different gene expression signatures.

Major comments:

The main caveat of the article is that the data is all based on bioinformatics associations without mechanistic studies, and there is no validation of the findings.

Since the recently published Grasso et al. Cancer Cell series did not make it to the datasets analyzed by the authors, they could consider using it as a validation set even though some samples overlap with the Riaz et al. Cell article (could be taken out for the validation analysis).

If the integration of the author's prediction with the IMPRES gene set is reported to improve the response and resistance prediction, why not propose the combined prediction as the main outcome of this article?

Minor comments:

The article needs careful review as it is plagued with typographical errors.

Reviewer #1, expert in network analysis (Remarks to the Author):

-- In this manuscript, the authors use a network-based approach to develop predictive methods that can be used identify genetic features, pathways, and therapeutic combinations associated with tumor response to anti-PD1 therapies. This is an interesting study that builds on other studies focused on molecular network-based analyses of complex disease mechanisms and therapeutic target/pathway identification.

We appreciate this reviewer's positive feedbacks and constructive critiques and we have now addressed all of reviewer's concerns below.

1). On page 3, the authors should point out that the tumor immune landscape also varies within a given tumor in an individual patient. This point also raises the important concern of sampling variability and its consequences for the soundness of statistical inference. Do the authors have access to (even limited) data sets from patients in whom repeated sampling was obtained, and to what extent does the MIAS score vary within such samples?

Answer:

- 1. Thanks for this valuable suggestion. We have now added a sentence in page 3 (also listed below) indicating that the tumor immune landscapes are heterogeneous in different regions within a tumor and relevant citation has also been included.**

Page 3: "Owing to the different genetic makeup of each tumor cell, the tumor immune landscape differs greatly not only between/within cancer types, but also varies in regions within a given tumor (Mitra et al., 2020)"

- 2. Among the datasets we have accessed, the Auslander (Auslander et al., 2018) and Abril-Rodriguez (Abril-Rodriguez et al., 2020; this dataset was just included in this revised manuscript as suggested by the reviewer 2) datasets have multiple samples from the same patients. We thus analyzed variations of MIAS scores and ESTIMATE immune infiltration scores (Yoshihara et al., 2013) of samples from the same patients in these two datasets. In the Auslander dataset, we found that the variations of MIAS scores of post-treated samples of patient 39 (7 samples) and 42 (8 samples) are big, compared to those of the post-treated samples of patient 62 (2 samples) and 208 (2 samples) (Figure R1). In addition, the variations of MIAS scores of post-treated samples from patient 39 and 42 are much bigger than those of pre-treated samples from the same patients. Similarly, we observed the same pattern in variations of ESTIMATE immune infiltration scores (Figure R2). In the Abril-Rodriguez dataset, MIAS and ESTIMATE immune infiltration scores of post-treated samples in some patients also showed big variations (Figure R3 and 4). We were not able to do the same analysis for the pre-treated samples because no multiple pre-treated samples from the same patients were available in the Abril-Rodriguez dataset. In summary, our analysis from these datasets is consistent with the notion that immune heterogeneity exists in different regions within a tumor. The immune heterogeneity may**

be associated with the genomic heterogeneity in the different samples from the same patient. Recently, the Mitra's studies (Mitra et al., 2020) also highlighted the links between marked levels of genomic and immune heterogeneity within the physical space of a tumor by performing immunogenomic analyses on 67 samples from a patient. Clearly, the mechanisms of the treatments that induce variations of immune scores in multiple samples of the same patients still needs to be elucidated in the future when large sample cohorts become available.

Figure R1: MIAS scores in pre-treated and post-treated samples of patients in the Auslander dataset.

Figure R2: ESTIMATE immune infiltration scores in pre-treated and post-treated samples of patients in the Auslander dataset

Figure R3: MIAS scores in the post-treated samples of patients in the Abril-Rodriguez dataset.

Figure R4: ESTIMATE immune infiltration scores in the post-treated samples of patients in the Abril-Rodriguez dataset

Reference:

1. Yoshihara K, et al. Inferring tumour purity and stromal and immune cell admixture from expression data. *Nat Commun.* 2013;4:2612.
2. Mitra A et al., Spatially resolved analyses link genomic and immune diversity and reveal unfavorable neutrophil activation in melanoma. *Nat Commun.* 2020;11(1):1839

2).The authors should identify the elements of their approach that account for the modest improvement over prior methods (IMPRES, TIDE, GEP), and do so quantitatively (rather than qualitatively, as on p. 12).

Answer:

- 1.** Before we do the comparative analysis, we need to first emphasize that the MIAS score of a sample is defined as the ssGSEA (single sample GSEA) enrichment score of the selected top 100 immune-positive genes for the sample. The major steps in generating an immune-positive signature for predicting patient response to anti-PD1 in a given cancer type_(detailed in the method section of the manuscript) are: 1). the network guilt-by-association method was first used to calculate functional relatedness of each gene with the MHC I pathway genes; 2). Because the gene network we used in the network association contained molecular interactions across different cellular types, not all of the top-scored genes are associated with the MHC I pathway in a given cancer type. Therefore, the MHC I-network association result was then integrated with the melanoma immune correlation data, which was calculated using TCGA melanoma transcriptomic data, to select 100 top MHC I-associated genes, whose expression are also significantly positively correlated with the ESTIMATE immune infiltration score (Yoshihara et al., 2013) to calculate the MIAS scores for response prediction of melanoma patients to anti-PD1. Therefore, the 100 signature genes of our MIAS approach were selected mainly based on the two major elements: a). their MHC I network association score (denoted MHC-I); b). the correlation of their expression with the ESITMATE immune score in a given cancer type (denoted ESTIMATE).
- 2.** We then describe the concepts of the three prior methods below. The IMPRES prediction is based on 15 pairwise transcriptomics relations between immune checkpoint genes, which can best separate low- and high-risk neuroblastoma patients. The TIDE prediction is both based on the T cell dysfunction gene signature (selected by the gene expression correlation with cytotoxic T lymphocyte infiltration score and patient survival) and the T cell exclusion gene signature (selected by the gene expression correlation with three Immunosuppressive cell types). The GEP prediction is based on the selected 18 inflammatory genes.
- 3.** To determine the importance of the two aforementioned elements, MHC-I and ESTIMATE in our MIAS prediction, we considered four scenarios that select signature genes based on the combinations of the two elements to calculate the ssGSEA enrichment scores (see the Methods) for predicting the response of all the complied 411 melanoma samples (note: we included the Abril-Rodriguez 2020 dataset in this revised manuscript as suggested by the reviewer 2; the total number of samples used for prediction evaluation of our MIAS approach is increased from 354 to 411). The first scenario, denoted “MHC-I + ESTIMATE”, is our MIAS approach, whose signature genes were selected based on both of the two elements. Thus, the signature genes of the MIAS approach are those top MHC I-associated genes that are specifically associated with immune infiltration in melanoma. The second scenario, denoted “MHC-I + nESTIMATE”, selected the top 100 genes only based on the MHC-I element. Accordingly, the 100 genes in this scenario may not be all associated with immune infiltration specifically in melanoma. The third scenario, denoted

“nMHC-I + ESTIMATE”, selected the 100 top immune-positive genes only by the ESTIMATE element. These 100 genes in this scenario thus may not be all specifically associated with the MHC-I pathway and response to anti-PD1. The fourth scenario, denoted “nMHC-I + nESTIMATE”, randomly selected 100 genes for prediction (i.e., neither of the two elements was used to select the genes). We next compared the prediction performance of these four scenarios with those of 3 prior methods, IMPRES, TIDE, and GEP, by the Wilcoxon-Mann-Whitney test (Figure R5). The result showed that the predictive performance of our MIAS approach (i.e., “MHC-I + ESTIMATE”) was significantly better than those of the three other scenarios, indicating both of the two elements are important to the prediction performance of our MIAS approach. We found that the third “nMHC-I + ESTIMATE” scenario has a comparable prediction performance to GEP, whose prediction is also based on signature genes that are only associated with immune infiltration level. Although the signatures of TIDE are also related to immune infiltration level, it has a better performance than the third scenario and GEP, probably due to its two different types of signatures that are respectively associated with dysfunction and exclusion of T cell infiltration. Nevertheless, the prediction performance of TIDE is still much worse than our MIAS approach (MHC-I + ESTIMATE). One explanation for the better performance of MIAS in anti-PD1 response prediction would be that its signature genes are those MHC-I associated genes that are associated with immune infiltration specifically in melanoma. In addition, as compared to MIAS, TIDE, and GEP, the prediction of IMPRES covers a very different aspect of immune response mechanisms, which considers pairwise relations between the expression of selected immune checkpoint genes. However, their 15 pairwise transcriptomics relations were selected based on the training data of regressing & progressing neuroblastoma but not melanoma patient samples, treated with anti-PD1. However, the tumor immunoregulation would be context dependent or tissue-specific (also revealed in our analysis of TCGA data on page.8-9). Thus, not all of their 15 pairwise transcriptomics relations are predictive to anti-PD1 response in melanoma. The predictive performance of MIAS for patient response to anti-PD1 was also better than IMPRES, probably owing to the fact that our method was specifically designed based on MHC I pathway-association and TCGA melanoma data.

4. In summary, these results indicate that the integration of two major elements (MHC-I and ESTIMATE) in our MIAS approach significantly improved selection of signature genes that are specifically associated with patient response to anti-PD1 in a given cancer type and outperformed TIDE, GEP and IMPRES. Thus, we also revised paragraphs in page 12-14 (also listed below) and a figure (Fig. 6C) to describe these results.

Page 12-14: “We next compared the performance of our approach with that of 3 other recently proposed methods, IMPRES [16], TIDE [17], and GEP [75], by the Wilcoxon-Mann-Whitney test (Fig. 6B) and AUC (Additional file 2: Fig. S5). The results showed that the predictive performance of our MIAS approach was better than those of the three prior methods in most of the datasets. The prediction of MIAS in melanoma was based on the top 100 immune-positive MHC I-associated genes that were selected mainly by the integration of two elements (see the equation 8 in the Methods): a). their MHC I association score (denoted MHC-I); b). the correlation of their

expression with the ESITMATE immune score in melanoma (denoted ESTIMATE). To determine how these two elements of the MIAS approach account for the prediction performance improvement over the three prior methods, we considered four scenarios that select signature genes based on the combinations of the two elements to calculate the ssGSEA enrichment scores (see the Methods) for predicting the response of all the complied 411 melanoma samples. The first scenario, denoted “MHC-I + ESTIMATE”, is our MIAS approach, whose signature genes were selected based on both of the two elements. Thus, the signature genes of the MIAS approach are those top MHC I associated genes that are specifically associated with immune infiltration in melanoma. The second scenario, denoted “MHC-I + nESTIMATE”, selected the top 100 genes only based on their MHC I association score. Accordingly, the 100 MHC I-associated genes in this scenario may not be all specifically associated with immune infiltration in melanoma. The third scenario, denoted “nMHC-I + ESTIMATE”, selected the top 100 immune-positive genes only based on the ESITMATE element. The 100 immune-associated genes in this scenario thus may not be all specifically associated with the MHC-I pathway and response to anti-PD1. The fourth scenario, denoted “nMHC-I + nESTIMATE”, randomly selected 100 genes for prediction (i.e., neither of the two elements was used to select the genes). We next compared the prediction performance of these four scenarios with those of 3 prior methods by the Wilcoxon-Mann-Whitney test (Fig. 6C). The result showed that the predictive performance of our MIAS approach (MHC-I + ESTIMATE) was significantly better than those of the three other scenarios, indicating both of the two elements are important to the prediction performance of MIAS. We also found that the third “nMHC-I + ESTIMATE” scenario has a comparable prediction performance to GEP. Similar to the third scenario whose signature genes are only associated with immune infiltration level, GEP’s prediction is based on expression level of 18 inflammatory genes [75]. Although the signatures used by TIDE in prediction are also related to immune infiltration level, it has a better performance than the third scenario and GEP, probably due to its two different types of signatures that are respectively associated with dysfunction and exclusion of T cell infiltration [17]. Nevertheless, the prediction performance of TIDE is still much worse than our MIAS approach. One explanation for the better performance of MIAS in anti-PD1 response prediction would be that its signature genes are those top MHC I-associated genes that are specifically associated with immune infiltration in melanoma.

In addition, as compared to MIAS, TIDE, and GEP, the prediction of IMPRES covers different aspects of immune response mechanisms. The IMPRES prediction is based on the 15 pairwise relations between the expression of immune checkpoint genes [16], which were selected using the training data of regressing & progressing neuroblastoma but not melanoma patient samples, treated with anti-PD1. Thus, the 15 pairwise transcriptomics relations may not all be predictive for anti-PD1 response in melanoma. Nevertheless, we reasoned that integration of the predictions of MIAS and IMPRES may improve the overall predictive performance because they would be complementary to each other. We evaluated the prediction performance of the integrated prediction for the combined dataset (e.g. all the 411 samples) using the Wilcoxon-Mann-Whitney test (Fig. 6D) and AUC (Additional file 2: Fig. S6). The result shows that the performance of the integrated prediction was better than that of the 2 individual approaches. In contrast, we did not see the performance was improved in the integrated prediction of MIAS and the other 2 prior methods.”

Figure R5: Performance comparison of all the model scenarios with three prior methods, IMPRES, TIDE, and GEP, using Wilcoxon test for the combined dataset (merged the datasets from all cohorts). The dotted line in the bar plots represents p-value = 0.05.

3). The notion of hubs as therapeutic targets (associated with anti-PD1 therapy) has significant potential shortcomings, given that by their highly connected nature, hub proteins are engaged in multiple other pathways that govern normal cell function. This important issue should be discussed.

Answer: We appreciate the insightful comment. We agree that hub genes are engaged in multiple other pathways that govern normal cell functions. Therefore, we should integrate our network prediction with other data, such as differential gene expression data between non-responders and responders, or between tumor and normal samples (Fig. 1B in the manuscript), or immune infiltration correlation data in a cancer type (Fig. 1C in the manuscript), to identify therapeutic targets that are specifically associated with anti-PD1 in cancer. In order to address this point, we added a sentence in “Cancer genes associated with response to anti-PD1 therapy” section of the discussion in page 16 (also listed below).

Page 16: “Our centrality analysis revealed that some cancer genes act like hubs in their molecular networks and are able to drive immune evasion through directly and indirectly deregulating the MHC I pathways. Therefore, some network centrality methods are also able to be used to identify genes and pathways associated with response and resistance to anti-PD1. The results of an in vivo study by Lesterhuis et al. [83] support this view. Lesterhuis and colleagues used analysis of gene networks inferred from gene expression data of responding and nonresponding tumors in murine models to identify hub genes and modules associated with response to anti-CTLA-4. They showed that targeting some of these identified hub genes with selected drugs dramatically enhanced the efficacy of CTLA-4 blockade in their murine models. However, the targets identified in mice may not be applicable in human tumors, and co-expression networks constructed from small samples may contain many false positives and negatives [84]. Nonetheless, this evidence shows that mutations or deregulation of some hub genes in the gene network can contribute to resistance to immunotherapies; importantly, many of these gene and their pathways are targetable. However, hub genes are normally engaged in multiple other pathways that govern normal cell functions. Thus, to identify hub gene as therapeutic targets that are specifically associated with anti-PD1 in cancer, the hub genes predicted by network approaches still need to be prioritized using other data, such as differential gene expression data between tumor and normal samples, or gene expression data of samples from cancer patients treated with anti-PD-1 therapy (Fig. 1B) or immune infiltration correlation data in cancer (Fig. 1C).”

4).The predictions of combination therapies that should enhance anti-PD1 therapy is most interesting and should be tested in vitro (at the very least), especially for agents that have not yet been in clinical trials.

Answer:

We appreciate this reviewer’s acknowledgement of our discovery work that proposed potential combination therapeutic strategy with anti-PD1 using our computational approach. Again, our prediction has been validated by the 155 targets of the 36 compounds (Table S5) that have been tested in clinical trials or used for combination treatments with anti-PD-1 therapies in cancer (Fig. 5). In addition, we also have identified several genes whose inhibitors recently have been showed clinical efficacy in combination with anti-PD1, such as HDAC (Suraweera et al., 2018), PARP (Jiao et al., 2017), GSK3B (Taylor et al., 2018), and PTK2 (Jiang et al., 2016). **In particular, we recently also found that several studies published in this year proposed some genes or their inhibitors that are associated with patient response to anti-PD1. These genes are also in the top list of our prediction (Table S1 in our manuscript).** First, Litchfield et al. (Litchfield et al, 2021) analyzed whole-exome and transcriptomic data for about 1,000 checkpoint inhibitor-treated patients (anti-CTLA-4, PD-1, or PDL1) collected from “12 published cohorts” across 7 tumor types, and identified two genes, TRAF2 (top 4.42% in our prediction) and CCND1 (top 6.5% in our prediction) to be associated with patient response. Second, SETDB1(top 15.13% in our prediction) was found to be associated with MHCI presentation and CD8+ T cell recognition of transposable element-encoded antigens (Griffin et al., 2021). Amplification of SETDB1 in human tumours is associated with immune exclusion and resistance to immune checkpoint blockade, and SETDB1 loss can

sensitizes tumours to anti-PD1 (Griffin et al., 2021). Third, an inhibitor of LGALS3 (top 8.78% in our prediction), belapectin, can enhance clinical and immunological effects of anti-PD1 (Curti et al., 2021). All these could be considered as the cross-validation results of our predictions. We will reference these new cross-validated targets in the discussion section of this revised manuscript (page14-15; also listed below).

We also agree with the reviewer that our prediction model should be further tested to identify novel agents that are not in clinical trials. If such agents are identified, this will significantly improve melanoma patients' survival given that the current immunotherapy is still limited to a subset of cancer patients. However, we are not aware of any in vitro assays to test the efficacy of combination therapy for patients. We have initiated both targeted approaches to test novel 36 druggable targets through CRISPR approach followed by drug combination testing in melanoma animal models. Furthermore, we also have begun the CRISPR screening approach on the predicted targets that are negatively correlated with the ESTIMATE immune infiltration score to further validate our computational prediction. However, the work outlined is labor intensive and time consuming, and we expect these experimental works will take up to 8-12 months. Given the aforementioned cross validation results that we found recently, we hope the reviewer will agree that our approach is sound and reasonable in this revised manuscript.

Reference:

1. Suraweera A, O'Byrne KJ, Richard DJ. Combination therapy with histone deacetylase inhibitors (HDACi) for the treatment of cancer: achieving the full therapeutic potential of HDACi. *Front Oncol.* 2018;8:92.
2. Jiao S, Xia W, Yamaguchi H, et al. PARP inhibitor upregulates PD-L1 expression and enhances cancer-associated immunosuppression. *Clin Cancer Res.* 2017;23(14):3711-20.
3. Taylor A, Rothstein D, Rudd CE. Small-molecule inhibition of PD-1 transcription is an effective alternative to antibody blockade in cancer therapy. *Cancer Res.* 2018;78(3):706-17.
4. Jiang H, Hegde S, Knolhoff BL, et al. Targeting focal adhesion kinase renders pancreatic cancers responsive to checkpoint immunotherapy. *Nat Med.* 2016;22(8):851-60.
5. Litchfield K. et al. Meta-analysis of tumor- and T cell-intrinsic mechanisms of sensitization to checkpoint inhibition. *Cell* 2021;184(3):596-614.e14.
6. Griffin GK, Wu J, Iracheta-Vellve A, Patti JC et al. Epigenetic silencing by SETDB1 suppresses tumour intrinsic immunogenicity. *Nature* 2021 May 5. PMID: 33953401
7. Curti BD et al. Enhancing clinical and immunological effects of anti-PD-1 with belapectin, a galectin-3 inhibitor. *J Immunother Cancer.* 2021;9(4):e002371.

Page14-15: "Our MHC I-association network approach successfully identified genes and pathways known to be associated with anti-PD1 response. The prediction was also comprehensively validated by several independent benchmark gene sets, including 6 CRISPR gene sets associated with tumor resistance to cytotoxic T cells and targets of the 36 compounds that have been tested in clinical trials for combination treatments with anti-PD1. In addition, target genes of several compounds that were recently shown to be able to enhance tumor response to anti-PD1 are also in top list of our prediction (Table S1 in our manuscript), such as

inhibitors of GSK3B, CDK, and PTK2. In particular, several very recent studies also cross-validated our prediction. First, Litchfield et al. [76] analyzed genomic data of up to 1,000 ICB-treated patient samples collected from 12 published cohorts across 7 tumor types, and showed that copy number loss of TRAF2 is associated with response and CCND1 amplification is associated with resistance. TRAF2 and CCND1 are respectively in top 4.42% and 6.5% in our prediction list. Second, SETDB1 (top 15.13% in our prediction) was found to be associated with MHC I presentation and CD8+ T cell recognition of transposable element-encoded antigens [77]. Amplification of SETDB1 in human tumours is associated with immune exclusion and resistance to immune checkpoint blockade, and SETDB1 loss can sensitize tumours to anti-PD1 [77]. Third, belaepectin, an inhibitor of LGALS3 (top 8.78% in our prediction), can enhance clinical and immunological effects of anti-PD1 [78]. All these demonstrated that our approach can effectively identify genes and pathways associated with response to anti-PD-1 therapy in cancer.”

5).The pre- and on-treatment response analysis is informative, but has been insufficiently analyzed. Specifically, in a validation set, one should be able to use the pre-treatment analysis to predict the on-treatment changes in gene expression, pathway modulation by treatment, and MIAS score.

Answer:

1. Our MIAS method was designed to predict patient response using the selected_MHC-I signature genes, but not to predict the on-treatment changes in gene expression and pathway modulation. From the results of immune heterogeneity analysis (question 1 of reviewer 1), we also found the treatments can induce variations of immune infiltration level in multiple samples of the same patients. Therefore, gene expression and pathway modulation may be very heterogeneous in on-treatment samples. Anyhow, we may need large sample cohorts that contain paired pre- and on-treatment samples from the same patients to do a deep analysis and understand to how to build up such as a predictive model.
2. We also need to emphasize that the anti-PD1 response prediction in melanoma of our MIAS approach was based on the 100 immune positive MHC-I signature genes (Figure 1C). The 100 signature genes were selected by integrating: a). our MHC I-network association prediction; b). the correlation data of gene expression and the ESTIAMTE immune infiltration scores across all TCGA melanoma samples. We used this signature to calculate the MIAS scores of the 411 pre- and on-treated melanoma samples compiled from 6 anti-PD1 cohorts, and found a good correlation with patient response to anti-PD1 for the 411 samples. **Therefore, none of these 411 melanoma samples was used to train or select the gene signature. Actually, these 411 samples (290 pre-treatment and 121 on-treatment) can be considered as the validation samples while the TCGA melanoma samples are the training samples for our prediction to anti-PD1**

response in melanoma (p.s., all the TCGA melanoma samples are all pre-treated). We also revised a paragraph in page 11-12 to avoid the confusion (also listed below).

Page 11-12: “Our MHC I-association prediction can be integrated with TCGA transcriptomic data of a given cancer type to select signature genes to calculate the MIAS score for predicting patient response to anti-PD1 therapies in the cancer type (Fig. 1C). In this work, we demonstrated this capability by applying our approach to melanoma, for which the most anti-PD-1 therapy cohorts are available. We used a meta-analysis method (see the Methods) to integrate our association prediction with the gene expression-immune correlation data, calculated using TCGA melanoma samples, to selected the 100 top immune-positive MHC I-associated signature genes (Additional file 1: Table S6) for response prediction to anti-PD-1 therapies for melanoma patients. Herein the TCGA melanoma samples can be considered as the training set for selecting the signature for prediction. We then validated the prediction power of this signature using 411 samples compiled from 6 melanoma cohorts (Additional file 1: Table S7) [16,68-72], in which patients were treated with anti-PD-1 therapy alone or in combination with anti-CTLA-4 therapy. We analyzed the gene expression data of the 411 samples and calculated their MIAS scores using this signature (see the Method). The MIAS scores and the clinical response data of the samples were then used to calculate the AUC to quantify the predictive performance of our MIAS approach. This evaluation was applied to each cohort dataset individually as well as the combined dataset that merged the datasets from all cohorts. Fig. 6A shows that the AUC values of most of the data sets were substantially higher than the random expectation (AUC=0.5). However, since size of some data sets are small, the ROC curve evaluation may not be reliable [73]. Thus, we also used the Wilcoxon-Mann-Whitney statistic, which is directly connected to the AUC of a ROC curve [74], to evaluate the performance of our predictions. Indeed, we found the results of the Wilcoxon tests and the AUC to be inconsistent in some small data sets; that is, some small data sets had very high AUC values but nonsignificant p-values on the Wilcoxon test (e.g., the Auslander.PD1.Pre_2018 data set).”

3. It seems that signature genes (selected from pre-treatment TCGA samples) of our MIAS approach have a better predictive performance for on-treated samples than for pre-treated samples (Fig. 6A). In order to understand more about the difference between pre- and on-treated samples, we also compared the composition of the immune infiltrated cells between all of the 411 pre- and on-treatment samples using ssGSEA enrichment scores of 29 immune cell gene signatures (Charoentong et al., 2017). We also found that most of the on-treatment samples tend to have more infiltrated cells than pre-treatment samples (Figure R6). This is consistent with the expectation that anti-PD1 can boost the immune response. As the aforementioned discussion in this rebuttal letter, the predictions of MIAS, TIDE, and GEP are based on signature genes associated with immune infiltration. Thus, the poor performances of these methods in pre-treatment datasets are probably due to the low immune infiltration level in these

samples.” We also revised a paragraph in page 14 to describe these results (also listed below).

Page 14: “Furthermore, we found MIAS and the three prior methods all performed poor in pre-treatment datasets, but performed significantly better in on-treatment datasets, except in the Gide data set [71]. To investigate the difference between the pre- and on-treatment samples, we compared their ESTIAMTE immune infiltration scores [26] and found that the immune infiltration level in pre-treatment datasets are all significantly lower than in on-treatment datasets (Additional file 2: Fig. S7). We also compared the composition of the immune infiltrated cells between all the pre- and on-treatment samples, calculated using ssGSEA enrichment scores of 29 immune cell gene signatures [15], and found that most of on-treatment samples tend to have more infiltrated immune cells than pre-treatment samples (Fig. S8). This is consistent with the expectation that anti-PD1 can boost the immune response. As the aforementioned paragraph , the predictions of MIAS, TIDE, and GEP are based on signature genes associated with immune infiltration. Thus, the poor performances of these methods in pre-treatment datasets are probably due to the low immune infiltration level in these samples. Other types of data, such as mutation burden, may need to be integrated for more robust response prediction in pre-treatment samples.”

Reference:

1. Charoentong, P., et al., Pan-cancer Immunogenomic Analyses Reveal Genotype-Immunophenotype Relationships and Predictors of Response to Checkpoint Blockade. *Cell Rep*, 2017. 18(1): p. 248-262.

Figure R6: Immune cell profile across all the pre- and on-treated melanoma samples compiled from the 6 anti-PD1 patient cohorts. Immune cell profiles were characterized using single-sample GSEA scores of immune cell gene sets.

6). The random walker approach to network proximity (‘guilt-by-association’) is one of several possible methods that can be used for the initial network analysis. The authors should consider (at least in the Discussion) some of the others that have been used (simple proximity, rather than ‘diffusion’ via the walker; neural network-based AI; etc.).

Answer: Thanks for this beneficial suggestion. We agree with the reviewer that other network approaches can also be used to identify genes and pathways associated with anti-PD1. Our network centrality analysis also showed that network centrality approaches also have ability for the prediction. We thus add a sentence in the page 19 (also listed below) to let reader understand more about the application of network approaches to identify genes/pathways associated with response/resistance to anti-PD1 therapy.

Page 19: “While our approach cannot be used to identify biomarker genes whose molecular interactions have not yet been well characterized because it is based on incomplete molecular interaction data, it will improve over time as more data are generated. Regardless, these results suggest that our network approach is an effective method to identify genes/pathways associated with response/resistance to anti-PD1 therapy and can be employed as in-silico screening of potential drugs for combination regimens with anti-PD1 therapy in cancer. In addition, some other network approaches, such as network centrality, direct neighborhood relationship, and shortest distance [19, 86], also can be used to identify genes and pathways associated with anti-PD1.”

Reviewer #2, expert in biomarkers/immunotherapies (Remarks to the Author):

This is an interesting article based on the re-analysis of publicly available sequencing data of biopsies of patients with cancer treated with immune checkpoint blockade therapies. The strength is the combined analysis of over 350 biopsies, allowing the comparison of different gene expression signatures.

We appreciate this reviewer's positive feedbacks and constructive critiques and we have now addressed all of reviewer's concerns below.

Major comments:

1).The main caveat of the article is that the data is all based on bioinformatics associations without mechanistic studies, and there is no validation of the findings.

Answer:

We need to emphasize our MHC I-association prediction was validated by 6 “independent” CRISPR gene sets associated with tumor resistance to cytotoxic T cells and targets of the 36 compounds that have been tested in clinical trials for combination treatments with anti-PD1. In addition, we also have identified several genes whose inhibitors were recently shown to have clinical efficacy in combination with anti-PD1, such as HDAC (Suraweera et al., 2018), PARP (Jiao et al., 2017), GSK3B (Taylor et al., 2018), and PTK2 (Jiang et al., 2016). **In particular, we recently also found that several studies published in this year proposed some genes or their inhibitors that are associated with patient response to anti-PD1. These genes are also in the top list of our prediction (Table S1 in our manuscript).** First, Litchfield et al. (Litchfield et al, 2021) analyzed whole-exome and transcriptomic data for about 1,000 checkpoint inhibitors (anti-CTLA-4, PD-1, or PDL1)-treated patient samples collected from “12 published cohorts” across 7 tumor types, and identified two genes, TRAF2 (top 4.42% in our prediction) and CCND1 (top 6.5% in our prediction) to be associated with patient response. Second, SETDB1 (top 15.13% in our prediction) was found to be associated with MHC I presentation and CD8+ T cell recognition of transposable element-encoded antigens (Griffin et al., 2021). Amplification of SETDB1 in human tumours is associated with immune exclusion and resistance to immune checkpoint blockade, and SETDB1 loss can sensitize tumours to anti-PD1 (Griffin et al., 2021). Third, an inhibitor of LGALS3 (top 8.78% in our prediction), belataceptin, can enhance clinical and immunological effects of anti-PD1 (Curti et al., 2021). All these could be considered as the cross-validation results of our predictions. We will reference these new cross-validated targets in this revised manuscript (Page 14-15; also listed below).

We appreciate this reviewer’s concern on the lack of mechanistic aspect of the work, which will require further experimental validation through unbiased CRISPR screening of our predicted targets in an animal model. This will take at least 12 months to address the mechanistic aspect

of the work following our identification and validation of hits in the animal model. This would significantly delay the communication of this important work to our colleagues in the field. We think the aforementioned new studies can cross validate our predictions well, and hope the reviewer will agree that our approach is sound and reasonable in this revised manuscript.

Page 14-15: “Our MHC I-association network approach successfully identified genes and pathways known to be associated with anti-PD1 response. The prediction was also comprehensively validated by several independent benchmark gene sets, including 6 CRISPR gene sets associated with tumor resistance to cytotoxic T cells and targets of the 36 compounds that have been tested in clinical trials for combination treatments with anti-PD1. In addition, target genes of several compounds that were recently shown to be able to enhance tumor response to anti-PD1 are also in top list of our prediction (Table S1 in our manuscript), such as inhibitors of GSK3B, CDK, and PTK2. In particular, several very recent studies also cross-validated our prediction. First, Litchfield et al. [76] analyzed genomic data of up to 1,000 ICB-treated patient samples collected from 12 published cohorts across 7 tumor types, and showed that copy number loss of TRAF2 is associated with response and CCND1 amplification is associated with resistance. TRAF2 and CCND1 are respectively in top 4.42% and 6.5% in our prediction list. Second, SETDB1(top 15.13% in our prediction) was found to be associated with MHC I presentation and CD8+ T cell recognition of transposable element-encoded antigens [77]. Amplification of SETDB1 in human tumours is associated with immune exclusion and resistance to immune checkpoint blockade, and SETDB1 loss can sensitizes tumours to anti-PD1 [77]. Third, belapectin, an inhibitor of LGALS3 (top 8.78% in our prediction), can enhance clinical and immunological effects of anti-PD1 [78]. All these demonstrated that our approach can effectively identify genes and pathways associated with response to anti-PD-1 therapy in cancer.”

Reference:

1. Suraweera A, O'Byrne KJ, Richard DJ. Combination therapy with histone deacetylase inhibitors (HDACi) for the treatment of cancer: achieving the full therapeutic potential of HDACi. *Front Oncol.* 2018;8:92.
2. Jiao S, Xia W, Yamaguchi H, et al. PARP inhibitor upregulates PD-L1 expression and enhances cancer-associated immunosuppression. *Clin Cancer Res.* 2017;23(14):3711-20.
3. Taylor A, Rothstein D, Rudd CE. Small-molecule inhibition of PD-1 transcription is an effective alternative to antibody blockade in cancer therapy. *Cancer Res.* 2018;78(3):706-17.
4. Jiang H, Hegde S, Knolhoff BL, et al. Targeting focal adhesion kinase renders pancreatic cancers responsive to checkpoint immunotherapy. *Nat Med.* 2016;22(8):851-60.
5. Litchfield K. et al. Meta-analysis of tumor- and T cell-intrinsic mechanisms of sensitization to checkpoint inhibition. *Cell* 2021;184(3):596-614.e14.
6. Griffin GK, Wu J, Iracheta-Vellve A, Patti JC et al. Epigenetic silencing by SETDB1 suppresses tumour intrinsic immunogenicity. *Nature* 2021 May 5. PMID: 33953401
7. Curti BD et al. Enhancing clinical and immunological effects of anti-PD-1 with belapectin, a galectin-3 inhibitor. *J Immunother Cancer.* 2021;9(4):e002371.

2). Since the recently published Grasso et al. Cancer Cell series did not make it to the datasets analyzed by the authors, they could consider using it as a validation set even though some samples overlap with the Riaz et al. Cell article (could be taken out for the validation analysis).

Answer:

1. We first need to emphasize that the anti-PD1 response prediction in melanoma of our MIAS approach was based on the 100 immune-positive MHC-I associated genes (Figure 1C). The 100 signature genes were selected by integrating (see the equation 8 in the Methods): a). our MHC I-network association prediction; b). the correlation data of gene expression and the ESTIAMTE immune infiltration scores across all TCGA melanoma samples. We used this signature to calculate the MIAS scores of the 411 pre- and on-treated melanoma samples compiled from 6 anti-PD1 cohorts (we just included the Abril-Rodriguez_2020 dataset in this revised manuscript as suggested by the reviewer; the total number of samples used for prediction evaluation of our MIAS approach is increased from 354 to 411), and found a good correlation between the MIAS score and patient response to anti-PD1. Therefore, none of these 411 melanoma samples was used to train or select the gene signature. Actually, these 411 samples can be considered as the validation samples while the TCGA melanoma samples are the training samples for our prediction to anti-PD1 response in melanoma. We also revised a paragraph in page 11-12 to avoid the confusion (also listed below).

Page 11-12: “Our MHC I-association prediction can be integrated with TCGA transcriptomic data of a given cancer type to select signature genes to calculate the MIAS score for predicting patient response to anti-PD1 therapies in the cancer type (Fig. 1C). In this work, we demonstrated this capability by applying our approach to melanoma, for which the most anti-PD-1 therapy cohorts are available. We used a meta-analysis method (see the Methods) to integrate our association prediction with the gene expression-immune correlation data, calculated using TCGA melanoma samples, to selected the 100 top immune-positive MHC I-associated signature genes (Additional file 1: Table S6) for response prediction to anti-PD-1 therapies for melanoma patients. Herein the TCGA melanoma samples can be considered as the training set for selecting the signature for prediction . We then validated the prediction power of this signature using 411 samples compiled from 6 melanoma cohorts (Additional file 1: Table S7) [16,68-72], in which patients were treated with anti-PD-1 therapy alone or in combination with anti-CTLA-4 therapy. We analyzed the gene expression data of the 411 samples and calculated their MIAS scores using this signature (see the Method). The MIAS scores and the clinical response data of the samples were then used to calculate the AUC to quantify the predictive performance of our MIAS approach. This evaluation was applied to each cohort dataset individually as well as the combined dataset that merged the datasets from all cohorts. Fig. 6A shows that the AUC values of most of the data sets were substantially higher than the random expectation (AUC=0.5). However, since size of some data sets are

small, the ROC curve evaluation may not be reliable [73]. Thus, we also used the Wilcoxon-Mann-Whitney statistic, which is directly connected to the AUC of a ROC curve [74], to evaluate the performance of our predictions. Indeed, we found the results of the Wilcoxon tests and the AUC to be inconsistent in some small data sets; that is, some small data sets had very high AUC values but nonsignificant p-values on the Wilcoxon test (e.g., the Auslander.PD1.Pre_2018 data set).”

2. We also agree with the reviewer’s suggestion to include new datasets for further validating our response prediction to anti-PD1. We found the Grasso dataset belongs to Bristol Myers Squibb and tried to contact them to get the permission to download the data. However, the communication about the clinical data transfer agreement between MD Anderson and Bristol Myers Squibb is still ongoing and consensus has not yet been reached by both parties on legal ground to finalize a Material Transfer Agreement (MTA). Fortunately, we found another new anti-PD1 dataset (Abril-Rodriguez et al., 2020). The result in Figure R7 shows that MIAS also have a good prediction power (AUC=0.864) on the on-treatment samples of the Abril-Rodriguez cohort, but a poor prediction power (AUC=0.45) on the pre-treatment samples. This result is consistent to those of other cohort samples (Fig. 6A). We now also include the result of this new dataset in the Fig. 6A. Now, we totally have 411 samples to validate our MIAS approach.

Reference:

1. Abril-Rodriguez G et al., PAK4 inhibition improves PD-1 blockade immunotherapy. *Nature Cancer* 2020; 1(1):1-13.

Figure R7 Performance of predictions of response to anti-PD-1 therapy (includes the result of the Abril-Rodriguez dataset).

3). If the integration of the author's prediction with the IMPRES gene set is reported to improve the response and resistance prediction, why not propose the combined prediction as the main outcome of this article?

Answer:

1. Thanks for this beneficial suggestion. In this manuscript, we mainly intend to demonstrate the capability of our network approach to identify genes and pathways associated with anti-PD1. We do understand the importance of integrative approaches in patient response prediction because several cancer cell-intrinsic and -extrinsic factors have been shown to be associated with response/resistance to anti-PD1. In the discussion section, we thus have several sentences addressing this point that we shall develop integrative computational approaches on the basis of all these factors to predict patient response (page.17-18; also listed below).
2. However, in order to help people to predict responses of SKCM patient samples directly using their transcriptomic data, we also used the MIAS and IMPRES scores of the collected

411 melanoma samples (290 pre-treatment and 121 on-treatment) as the data features and applied support vector machine (SVM) to trained integrative anti-PD1 response predictors for pre- and on-treatment melanoma samples. The R script of using these predictors is available in the GitHub repository (<https://github.com/perwu/MIAS>). We also added a paragraph in page 14 to address this point (also listed below).

Page 17-18: “Integration of our MHC I-association prediction with TCGA transcriptomic data of a given cancer type can select signature genes for calculating the MIAS score to predict patient response to anti-PD1 of the cancer type. In this work, we demonstrated this capability in melanoma. We selected the signature genes (e.g. top immune-positive MHC I-associated genes) in melanoma using TCGA SKCM transcriptomic data, and used the 411 melanoma samples compiled from 6 cohorts to validate the prediction performance of the MIAS score calculated by the signature genes. Our MIAS approach also could be applied to other cancer types to select the signature genes for predicting patient response in these cancer types. However, factors in addition to the deregulation of the MHC I pathway, which have been shown to be associated with response/resistance to anti-PD1, such as mutation burden, the microbiome, environmental factors, germline genetics, and immune infiltration, also need to be included in the response prediction [84]. Thus, development of integrative computational approaches on the basis of different cancer cell-intrinsic and -extrinsic factors would perform better than approaches that only consider an individual factor. For instance, the combination of mutation burden and tumor aneuploidy scores was shown to better predict response to checkpoint immunotherapies than either score alone [11, 12]. Our analysis also showed that the integration of the MIAS and IMPRES scores also have a better prediction performance than the two individual methods.”

Page 14: “All the aforementioned results indicate that MIAS score can be a useful feature to build integrative machine-learning models for response prediction of anti-PD1. The R script for calculating MIAS scores of melanoma samples is available in the GitHub repository (<https://github.com/perwu/MIAS>). We also used the MIAS and IMPRES scores of the collected 411 melanoma samples (290 pre-treatment and 121 on-treatment) as the data features and applied support vector machine (SVM) to trained anti-PD1 response predictors for pre- and on-treatment melanoma patient samples. The accuracy rates of these predictors calculated by the 5-fold cross validation were listed in Table S8. These predictors (also available at <https://github.com/perwu/MIAS>) can help people to predict responses of SKCM patient samples directly using their transcriptomic data.”

Minor comments:

The article needs careful review as it is plagued with typographical errors.

Answer: We apologize for the typos and have made corrections

Reviewers' Comments:

Reviewer #1:

Remarks to the Author:

N/A

Reviewer #2:

Remarks to the Author:

The authors have correctly addressed the major issues from the initial submission.

REVIEWERS' COMMENTS

1. Reviewer #1 (Remarks to the Author):

N/A

Answer: No action needed

2. Reviewer #2 (Remarks to the Author):

The authors have correctly addressed the major issues from the initial submission.

Answer: No action needed